# Disulfide-activated protein kinase G Iα regulates cardiac diastolic relaxation and fine-tunes the Frank–Starling response

Jenna Scotcher[1,*], Oleksandra Prysyazhna[1,*], Andrii Boguslavskyi[1], Kornel Kistamas[2], Natasha Hadgraft[3], Eva D. Martin[1], Jenny Worthington[4], Olena Rudyk[1], Pedro Rodriguez Cutillas[5], Friederike Cuello[6], Michael J. Shattock[1], Michael S. Marber[1], Maria R. Conte[7], Adam Greenstein[2], David J. Greensmith[3], Luigi Venetucci[2], John F. Timms[4] & Philip Eaton[1]

The Frank–Starling mechanism allows the amount of blood entering the heart from the veins to be precisely matched with the amount pumped out to the arterial circulation. As the heart fills with blood during diastole, the myocardium is stretched and oxidants are produced. Here we show that protein kinase G Iα (PKGIα) is oxidant-activated during stretch and this form of the kinase selectively phosphorylates cardiac phospholamban Ser16—a site important for diastolic relaxation. We find that hearts of Cys42Ser PKGIα knock-in (KI) mice, which are resistant to PKGIα oxidation, have diastolic dysfunction and a diminished ability to couple ventricular filling with cardiac output on a beat-to-beat basis. Intracellular calcium dynamics of ventricular myocytes isolated from KI hearts are altered in a manner consistent with impaired relaxation and contractile function. We conclude that oxidation of PKGIα during myocardial stretch is crucial for diastolic relaxation and fine-tunes the Frank–Starling response.

[1] King's College London, Cardiovascular Division, The British Heart Foundation Centre of Excellence, The Rayne Institute, St Thomas' Hospital, London SE1 7EH, UK. [2] Institute of Cardiovascular Sciences, Manchester Academic Health Science Centre, Core Technology Facility, University of Manchester, 46 Grafton Street, Manchester M13 9NT, UK. [3] Biomedical Research Centre, University of Salford, Peel Building, Salford Crescent M5 4WT, UK. [4] Institute for Women's Health, University College London, Gower Street, London WC1E 6BT, UK. [5] Barts Cancer Institute, Barts and The London School of Medicine and Dentistry, Queen Mary University of London, Charterhouse Square, London EC1M 6BQ, UK. [6] Department of Experimental Pharmacology and Toxicology, Cardiovascular Research Centre, University Medical Center Hamburg-Eppendorf, Hamburg, DZHK (German Centre for Cardiovascular Research), Partner site Hamburg/Kiel/Lübeck, Hamburg 20246, Germany. [7] Randall Division of Cell and Molecular Biophysics, King's College London, New Hunt's House, Guy's Campus, London SE1 1UL, UK. * These authors contributed equally to this work. Correspondence and requests for materials should be addressed to P.E. (email: philip.eaton@kcl.ac.uk).

Protein kinase G Iα (PKGIα) can be activated via the classical NO/cyclic guanosine monophosphate (cGMP) pathway or via a cGMP-independent pathway involving oxidants[1,2]. Reactive oxygen species (ROS) promote formation of a reversible intermolecular disulfide bond between the two subunits of the PKGIα homodimer at Cys42 (refs 3,4). This redox mechanism operates in blood vessels to control vasotone and blood pressure in vivo[5,6]. However, PKGIα is also expressed in heart muscle where the significance of Cys42 oxidation is less clear.

The amount of blood that enters the heart is continuously changing, for example with postural alterations and breathing. As the heart fills with blood during diastolic relaxation, myocardial cells become stretched. The magnitude of stretch is proportional to the volume of blood that enters the ventricle, and the force of the subsequent contraction is proportional to the degree of stretch. This mechanism, which ensures that the amount of blood pumped out (stroke volume) is synchronized with the amount that enters (venous return), is known as the Frank–Starling or Maestrini law of the heart[7–10]. This mechanism also enables beat-to-beat matching of left ventricular to right ventricular output.

Widely accepted molecular mechanisms that contribute to the Frank–Starling response include stretch-induced alterations in myofilament overlap, myofilament $Ca^{2+}$ sensitivity, and actin-myosin cross-bridge formation[11,12]. Recently, it was discovered that there is an increased production of ROS during diastolic stretch, termed X-ROS signalling, which is involved in regulation of cardiac $Ca^{2+}$ cycling[13].

Removal of cytosolic $Ca^{2+}$ to trigger diastolic relaxation occurs predominately via the sarcoplasmic reticulum (SR) $Ca^{2+}$ ATPase 2a (SERCA2a), which transfers $Ca^{2+}$ into the lumen of the SR to be stored before the next contraction[14]. SERCA2a activity is regulated via interactions with its reversible inhibitor phospholamban (PLN); when PLN is phosphorylated at Ser16, its inhibitory action on SERCA2a is relieved and $Ca^{2+}$ sequestration into the SR is increased. Myocardial relaxation is potentiated which enhances filling of the heart[15,16].

Here we identify PLN Ser16 as a direct target of disulfide PKGIα in the heart by an unbiased chemical genetic phosphoproteomic experiment utilizing analogue-sensitive PKGIα mutants[17]. We investigate the functional significance of disulfide PKGIα-dependent phosphorylation of PLN using a Cys42Ser PKGIα knock-in (KI) transgenic mouse and find that PKGIα oxidation occurs during stretch to contribute to the Frank–Starling response and is a key determinant of cardiac output.

## Results

### Disulfide-activated PKGIα phosphorylates PLN at Ser16.
A chemical genetic phosphoproteomic method utilizing analogue-sensitive kinase mutants was performed to identify direct cardiac substrates of PKGIα (ref. 18). Phosphopeptide abundance—directly relating to the amount of substrate phosphorylation—was determined by a label-free quantitative analysis[19,20] (Supplementary Dataset 1). Substrate phosphorylation was assessed when PKGIα was activated by the Cys42 disulfide bond or via the classical pathway with cGMP, and compared with phosphorylation when PKGIα was in its basal 'unactivated' state (control; Fig. 1a). Eighty five direct substrates of PKGIα were identified and the 29 substrates that had a statistically significant change in phosphorylation upon activation of the kinase are listed in Table 1. Phosphorylation of 28 of these proteins was significantly increased when PKGIα was activated by cGMP. Intriguingly, the abundance of a phosphopeptide from the cardiac protein phospholamban (PLN pSer16; RApSTIEMPQQAR) was

significantly increased relative to control when PKGIα was activated by Cys42 oxidation rather than by cyclic nucleotide binding, indicating that PLN Ser16 is a selective target of disulfide dimer PKGIα.

We compared basal phosphorylation of PLN Ser16 in isolated, buffer-perfused hearts from C42S PKGIα KI mutant mice (which cannot form the activating intermolecular disulfide bond) to Ser16 phosphorylation in wild-type (WT) hearts (Fig. 1b). PLN Ser16 phosphorylation was significantly lower in the KI hearts, consistent with the proteomic evidence that PLN is a substrate of disulfide-activated PKGIα. We observed no change in phosphorylation of PLN Thr17 in the KI tissue suggesting that disulfide PKGIα is highly selective for Ser16. As well as Ser16 phosphorylation, the oligomeric state of PLN was altered in the myocardium of the KI, as indicated by a three-fold increase in the pentamer/monomer ratio of total PLN in samples that were not boiled before western blotting.

Given that PLN plays a central role in cardiac excitation-contraction (EC) coupling and $Ca^{2+}$ homoeostasis, we analysed several other key proteins involved in these processes to determine whether their expression or phosphorylation status was altered in the KI myocardium (Fig. 1c). However, we observed no changes for any of the indices measured, including cardiac troponin I (cTnI) Ser22/23, cardiac myosin binding protein C (cMyBP-C) Ser282, ryanodine receptor 2 (RyR2) Ser2808, phospholemman (FXYD1) Ser63, Ser68 and Ser69, myosin light chain 2 (MLC2) Ser19, heavy chain cardiac myosin, slow myosin heavy chain and $Ca^{2+}$/calmodulin-dependent protein kinase II (CaMK2-β/γ/δ) Thr282.

### Impaired Frank–Starling mechanism in C42S PKGIα hearts.
To investigate the functional significance of oxidized PKGIα-dependent phosphorylation of PLN, we began by assessing the Frank–Starling relationship of perfused ex vivo hearts from WT or C42S PKGIα KI mice. The systolic pressure (SP), rate of contraction ($+dp/dt$), and rate of relaxation ($-dp/dt$), were monitored as the end-diastolic pressure (EDP), that is, cardiac preload, was sequentially increased. The KI hearts displayed a markedly different Frank–Starling profile from that of the WT hearts (Fig. 2a). A statistically significant elevation of EDP was required for the KI hearts to achieve the same SP as the WTs. For example, at an EDP of 4 mm Hg the WT heart generated a SP of ∼60 mm Hg, whereas the KI only generated a SP of ∼30 mm Hg. Furthermore, the KI hearts had significantly slower rates of contraction and relaxation than the WTs at a given EDP.

### Effect of stretch on PLN Ser16 phosphorylation.
We assessed the effect that EDP had on PKGIα oxidation state and PLN Ser16 phosphorylation in WT and KI ex vivo hearts by Western blot. Increasing EDP from 0 mm Hg to 5 mm Hg, thus increasing diastolic stretch, significantly increased oxidation of PKGIα to the disulfide dimer in WT hearts (Fig. 2b). As expected, this oxidation event was absent in hearts that harboured the PKGIα C42S mutation. An increase in EDP was also associated with a significant elevation in PLN Ser16 phosphorylation in WT myocardium, whereas Ser16 phosphorylation in the C42S mutant tissue was unchanged (Fig. 2c).

Subcellular fractionation of myocardial tissue from the WT and KI mice was also performed to see if increased stretch was associated with translocation of PKGIα. Indeed, we observed a statistically significant increase in the amount of WT PKGIα in the particulate fraction—where the SR is enriched and PLN and SERCA2a are located (Fig. 2d). However, the amount of C42S PKGIα in the SR-enriched fraction from KI heart tissue did not change. This observation is consistent with the

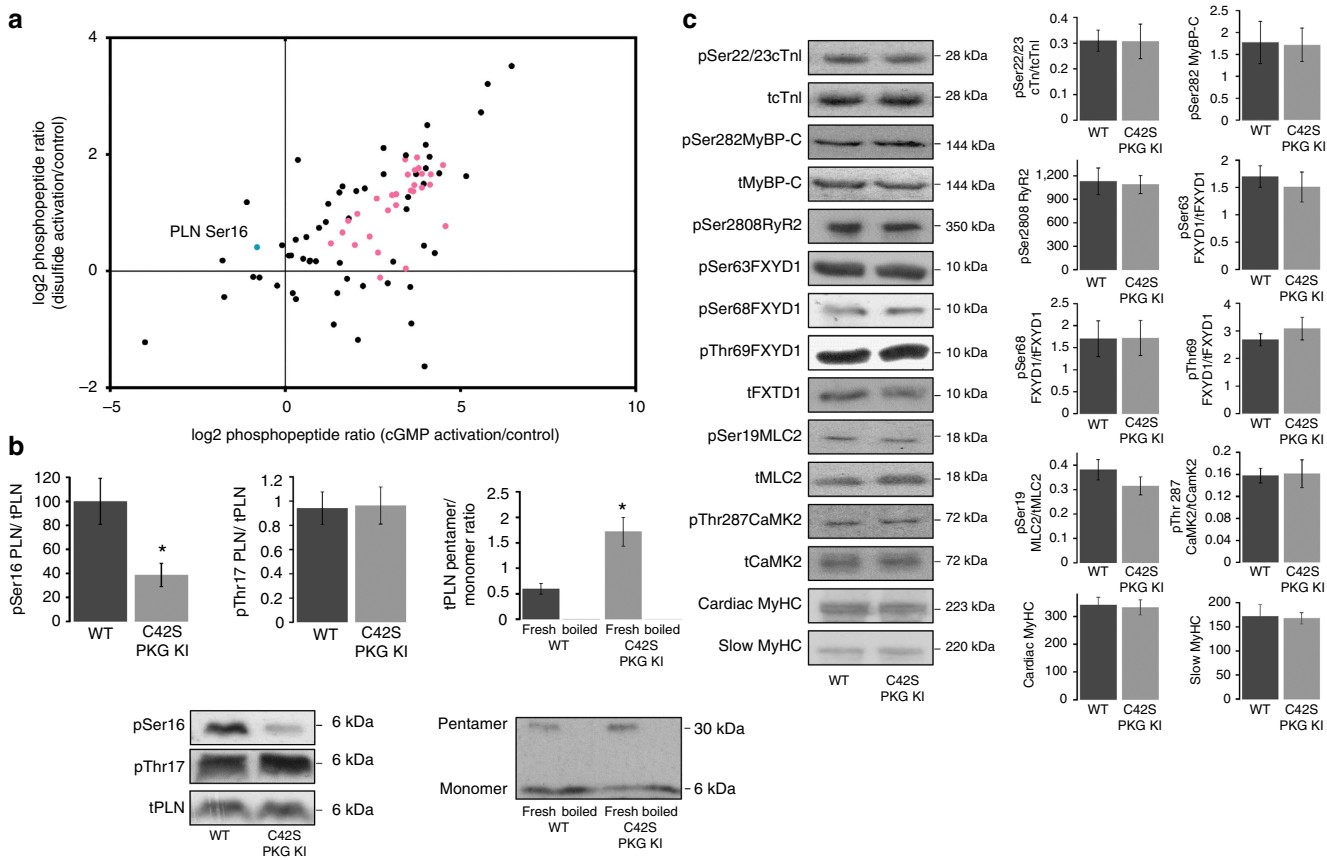

**Figure 1 | Identification of PLN Ser16 as a selective target of disulfide-activated PKGIα.** (**a**) Cardiac substrates of PKGIα were identified by a chemical genetic phosphoproteomic method and the amount of substrate phosphorylation was quantitated when PKGIα was in its basal state (control), activated by disulfide bond, or activated by cGMP. The scatter plot shows the PKGIα activation-dependent $\log_2$ fold changes in phosphopeptide abundance relative to control. Pink circles represent phosphopeptides whose abundances were significantly increased compared with control ($P < 0.05$; $n = 4$) when PKGIα activity was stimulated by cGMP and the blue circle represents a phosphopeptide whose abundance was significantly increased relative to control ($P < 0.05$; $n = 4$) when PKGIα was activated by disulfide; this phosphopeptide, RAS(p)TIEMPQQAR, is from the cardiac SR protein phospholamban (PLN) and is phosphorylated at residue Ser16. Decreases in phosphorylation were not statistically significant for any substrate. $P$ values were determined by post hoc Dunnett's test following one-way ANOVA. (**b**) Oxidized PKGIα-mediated phosphorylation of PLN was confirmed by a study of the basal level of Ser16 phosphorylation in isolated hearts from C42S PKGIα KI mice which cannot form the activating disulfide bond. PLN Ser16 phosphorylation was significantly decreased in the KI compared with WT (*$P < 0.05$; $n = 6$) while phosphorylation of Thr17 was unchanged. We also observed a significant three-fold increase in the pentamer/monomer ratio of total PLN in the KI hearts, indicating that the oligomeric state of PLN is also affected by the oxidation state of PKGIα. (**c**) Immunoblots from WT or KI myocardium for several other key proteins and their phosphosites involved in cardiac EC coupling and $Ca^{2+}$ handling. No significant changes in phosphorylation or total protein levels were detected between genotypes. Histograms show the mean ± s.e.m. and $P$ values were determined by $t$-test.

phosphoproteomic data which revealed that disulfide PKGIα directly interacts with PLN.

**Oxidized PKGIα binds to the cytoplasmic domain of PLN.** To explore the PLN-PKGIα interaction further we carried out isothermal titration calorimetry (ITC), titrating the cytoplasmic domain of PLN (residues 1–23; $PLN_{1-23}$) against the oxidized (WT) and reduced (C42S mutant) forms of PKGIα. A sigmoidal binding isotherm was fitted to the integrated titration data for oxidized PKGIα which is consistent with one PKGIα disulfide dimer binding to one PLN peptide with a $K_d$ of ~7 μM (Fig. 2e). In contrast, integrated heats for the mutant kinase, recorded under the same experimental conditions, could not be fitted to a sigmoid-shaped binding curve, therefore a dissociation constant for the C42S PKGIα-$PLN_{1-23}$ complex could not be derived from our experiments. Although the ITC data here does not exclude the possibility of an interaction between mutant PKGIα and PLN, it does suggest that the interaction between reduced, unactivated PKGIα and PLN is markedly weaker than the interaction between

oxidant-activated PKGIα and PLN. Using the MicroCal isotherm simulation tool, we estimated that the $K_d$ for the mutant kinase is at least five-fold higher than the $K_d$ for WT PKGIα disulfide dimer.

**$Ca^{2+}$ handling in myocytes from C42S PKGIα KI hearts.** Experiments were performed in ventricular myocytes isolated from adult WT or C42S PKGIα KI hearts, comparing intracellular calcium ($[Ca^{2+}]_i$) dynamics between genotypes. Specimen transients (Fig. 3a) are clearly consistent with significantly altered $Ca^{2+}$ handling in the cells from KI animals. Quantitative analysis of the transients showed the KI was significantly deficient in their systolic $[Ca^{2+}]_i$ transient and SR $Ca^{2+}$ content evoked by application of caffeine, whereas the diastolic $[Ca^{2+}]_i$ concentration was the same between genotypes (Fig. 3b–d). Normalization of the $[Ca^{2+}]_i$ transients allowed direct comparison of their decay phase (indicative of SERCA2a activity) between genotypes (Fig. 3e). The dashed lines show single exponential fits which were used to determine the rate constants for the decay of

**Table 1 | Direct substrates of PKGIα identified by a quantitative phosphoproteomic screen.**

| UniProt ID | Protein | Phospho site | Log$_2$ fold change | |
|---|---|---|---|---|
| | | | cGMP versus control | Disulfide versus control |
| P61016 | Cardiac phospholamban | S16 | − 0.81 | 0.41* |
| P10686 | 1-phosphatidylinositol 4,5-bisphosphate phosphodiesterase gamma-1 | S1233 | 4.50* | 1.82 |
| P30835 | 6-phosphofructokinase, liver type | S775 | 4.13* | 1.48 |
| Q99068 | Alpha-2-macroglobulin receptor-associated protein | S245 | 2.64* | 0.32 |
| P27653 | C-1-tetrahydrofolate synthase, cytoplasmic | T545 | 3.68* | 1.48 |
| O55156 | CAP-Gly domain-containing linker protein 2 | S353 | 2.05* | 0.98 |
| Q99JD4 | CLIP-associating protein 2 | S436 | 1.79* | 0.86 |
| Q9QXU8 | Cytoplasmic dynein 1 light intermediate chain 1 | T408 | 3.76* | 1.95 |
| Q9QXU8 | Cytoplasmic dynein 1 light intermediate chain 1 | S412 | 4.15* | 1.66 |
| F1LP64 | E3 ubiquitin-protein ligase TRIP12 | S1073 | 1.63* | 0.66 |
| Q9R080 | G-protein-signaling modulator 1 | S567 | 2.93* | 1.04 |
| P97541 | Heat shock protein beta-6 | S16 | 2.61* | 1.25 |
| P15865 | Histone H1.4 | S36 | 1.30* | 0.48 |
| D3ZBN0 | Histone H1.5 | T35 | 3.90* | 1.43 |
| P62804 | Histone H4 | S48 | 2.70* | − 0.11 |
| Q5SGE0 | Leucine-rich PPR motif-containing protein, mitochondrial | S656 | 4.57* | 0.77 |
| P43244 | Matrin-3 | T150 | 3.58* | 1.38 |
| P34926 | Microtubule-associated protein 1A | S460 | 2.41* | 0.59 |
| Q5M7W5 | Microtubule-associated protein 4 | T899 | 3.69* | 1.73 |
| P19332 | Microtubule-associated protein tau | S525 | 3.90* | 1.67 |
| Q5U2R4 | Mitochondrial ribonuclease P protein 1 | T377 | 1.98* | 0.45 |
| E9PT87 | Myosin light chain kinase 3 | S155 | 3.16* | 1.13 |
| P18437 | Non-histone chromosomal protein HMG-17 | S29 | 3.65* | 1.37 |
| P85125 | Polymerase I and transcript release factor | T304 | 3.05* | 1.30 |
| P85125 | Polymerase I and transcript release factor | S302 | 3.16* | 1.32 |
| Q8VBU2 | Protein NDRG2 | S332 | 3.42* | 1.91 |
| Q63945 | Protein SET | S7 | 3.80* | 1.76 |
| Q60587 | Trifunctional enzyme subunit beta, mitochondrial | S198 | 3.44* | 0.04 |
| P23693 | Troponin I, cardiac muscle | S167 | 3.49* | 1.65 |

All proteins that displayed a statistically significant log2 fold change in phosphorylation upon PKGIα activation by cGMP or Cys42 disulfide bond are listed (*$P < 0.05$, Dunnett's test; $n = 4$).

$[Ca^{2+}]_i$, which was significantly slower by $\sim 50\%$ in cells from KI mice (Fig. 3f).

**Diastolic relaxation in C42S PKGIα mutant mice *in vivo*.** We performed a comprehensive echocardiography study to assess the myocardial contractile function of WT and KI mice *in vivo*. The complete list of measurements is given in Supplementary Table 1. Importantly, the KI, which is resistant to PKGIα oxidative activation, had a significant decrease in the transmitral early ($E$) to late ($A$) peak flow velocity wave ratio compared with WT, indicating impaired diastolic relaxation (Fig. 4a)[21]. The mitral annulus early diastole tissue motion ($E'$) to mitral annulus late diastole tissue motion ($A'$) wave ratio was also decreased in the KI, providing further evidence for abnormal myocardial relaxation when the Cys42 PKGIα disulfide bond cannot form.

*In vivo* cardiac performance was further assessed by analysis of pressure-volume (PV) loops obtained from a catheter inserted in the left ventricle (LV) of the WT and KI mice; the complete list of measurements can be found in Supplementary Table 2. The KI hearts had significantly increased EDPs and were slower in both their contraction and relaxation rates (Fig. 4b). Furthermore, the reduced end-systolic pressure-volume relationship in the KI reveals a deficiency in contractile performance, and the elevated end-diastolic pressure-volume relationship indicates that the ventricle of the KI is stiffer. Compelling evidence for diastolic impairment in the KI is further provided by an elevated isovolumic relaxation constant, Tau, which is a preload-independent measure of diastolic function; a higher value indicates slower relaxation[22].

**The Frank–Starling mechanism in C42S PKGIα mice *in vivo*.** Based on the data described so far, we concluded that oxidation of PKGIα is an important mechanism for obtaining the appropriate degree of filling during the relaxation phase of the cardiac cycle. According to the Frank–Starling law—which couples end-diastolic volume (EDV) to cardiac output on a beat-to-beat basis—impaired filling in the KI should attenuate the force of the following contraction. To test this hypothesis, we varied the preload (EDV) of WT and KI mice by mechanical occlusion of the vena cava and recorded high-resolution PV data. Consequently, EDP, EDV, SP, $+ dp/dt$ and $- dp/dt$ could be determined for individual cardiac cycles as indicated in the representative trace in Fig. 4c. Thus, intra-beat relationships could be calculated, for example EDP versus SP, that pertained directly to the Frank–Starling response *in vivo*.

A representative scatter plot of SP versus EDP for an individual WT and KI mouse is shown in Fig. 4d. Two hundred heartbeats were analysed for each mouse and slopes were averaged in order to obtain the mean intra-beat relationships for EDP versus SP, EDP versus the rate of contraction, and EDP versus the rate of relaxation. On average there was a $\sim 14$ mm Hg increase in SP for a 1 mm Hg change in EDP in WT, while there was only a $\sim 10$ mm Hg increase in SP per unit change in EDP in the KI. The intra-beat relationships between EDP and $+ dp/dt$ and $- dp/dt$ were also significantly decreased in the KI. Additionally, the spread of data was strikingly greater in the scatter plots of EDP versus SP for the KI mice. Hence, we calculated the mean coefficient of determination ($R^2$) for each intra-beat relationship to assess quantitatively the variability of the data, thus providing a measure of how tightly linked EDP and cardiac output were in the KI mice compared with WT. $R^2$ for each relationship was

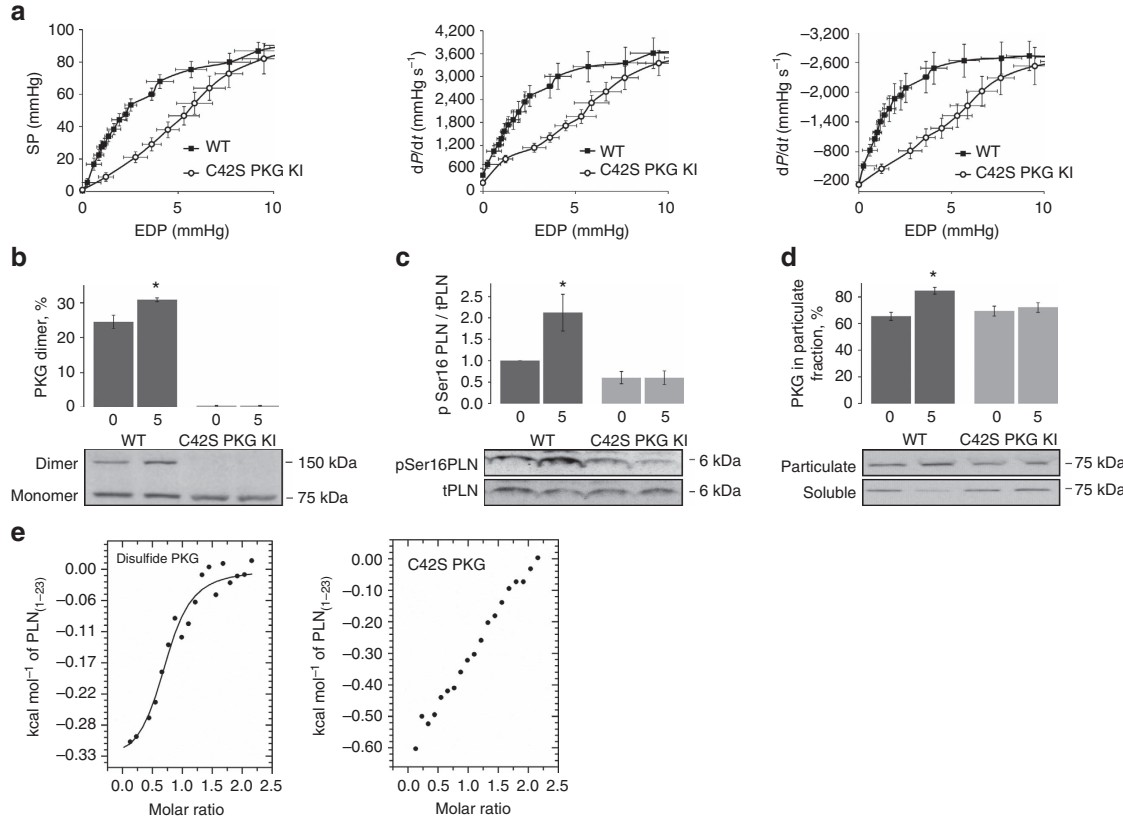

**Figure 2 | Isolated hearts from C42S PKGIα KI mice have impaired Frank–Starling responses.** (**a**) Curves showing the variation in SP, rate of contraction ($+\,dp/dt$), and rate of relaxation ($-\,dp/dt$) as a function of EDP for Langendorff-perfused WT and KI hearts. Cardiac performance increased with EDP, that is, stretch, according to the Frank–Starling law. However, the responses were significantly reduced in the KI hearts compared with WT ($P < 0.05$; $n = 8$). (**b**) Immunoblotting showed that oxidation of PKGIα to the disulfide dimer increased with increasing stretch (from 0 mm Hg to 5 mm Hg EDP) in the WT hearts but not in the KI (*$P < 0.05$; $n = 5$). (**c**) Phosphorylation of PLN Ser16 was also significantly increased in the WT but not KI hearts with increased diastolic stretch (*$P < 0.05$; $n = 5$). (**d**) Subcellular fractionation of WT and KI hearts perfused at different EDPs followed by immunoblotting revealed a significant increase in the amount of PKGIα in the particulate fraction from stretched WT myocardium compared with the particulate fraction from unstretched WT myocardium (*$P < 0.05$; $n = 5$). No stretch-dependent changes in PKGIα abundance were observed in fractions from the KI tissue. Error bars show s.e.m. and $P$ values were determined by $t$-test. (**e**) ITC analysis of the interaction between the cytoplasmic domain of PLN (residues 1–23) with disulfide-activated PKGIα and the C42S mutant. A sigmoidal binding isotherm can be fitted to the titration data for oxidized PKGIα which is consistent with one PKGIα disulfide dimer binding to one PLN peptide with a $K_d$ of $\sim 7\,\mu$M. Although the ITC data for C42S PKGIα also suggests a direct interaction with the cytoplasmic domain of PLN, this is markedly weaker than for oxidized PKGIα as the integrated data cannot be fitted to a sigmoid-shaped binding curve under the same experimental conditions.

significantly reduced in the KI mice, indicating that coupling between EDP and contractile function was distinctly impaired in the hearts of C42S PKGIα KI mice.

The intra-beat relationships for EDV versus SP, $-\,dP/dt$ and $+\,dP/dt$ were also determined and the findings mirrored those reported for EDP (Fig. 4e). Namely, the KI hearts did not develop as much SP as WT hearts per unit change in EDV and the positive relationships between EDV and the relaxation and contraction rates were diminished. Additionally, the data were much more variable as illustrated by significantly decreased $R^2$ values. EDV provides a surrogate index of myocardial stretch during diastole; thus the intra-beat relationships of SP and $+\,dP/dt$ versus EDV can be used as an *in vivo* readout of the Frank–Starling response. This data therefore provides further evidence that the C42S PKGIα KI mouse has a robust impairment in its ability to fully invoke the Frank–Starling mechanism.

**Force generated by the heart in C42S PKGIα KI mice.** We hypothesized that disconnection between diastolic filling and cardiac output in the KI would result in a more variable

developed pressure (that is SP − EDP). Thus, we derived the variance in the developed pressure amplitude over 1,000 consecutive heartbeats for mice of each genotype from PV data obtained with a catheter inserted in the LV. Indeed, the developed pressure—the force generated with each heartbeat—was ∼three-fold more variable in the KI (Fig. 4f). We performed a similar analysis on radiotelemetry data collected from conscious, freely moving WT and KI mice that had a pressure catheter inserted into the aorta. Similar to the data obtained with the PV catheter, the aortic pulse pressure of the KI was ∼2.2-fold more variable than that of the WT. These results further corroborate the importance of the PKGIα redox control mechanism in regulation of cardiac output.

## Discussion

PKGIα activity can be stimulated by reversible oxidation of Cys42 or by cGMP binding[1,2]. cGMP inhibits formation of the Cys42 intermolecular disulfide bond[3,4] and, similarly, oxidation has been shown to attenuate cGMP-dependent PKGIα substrate phosphorylation[23], consistent with discrete PKGIα signalling

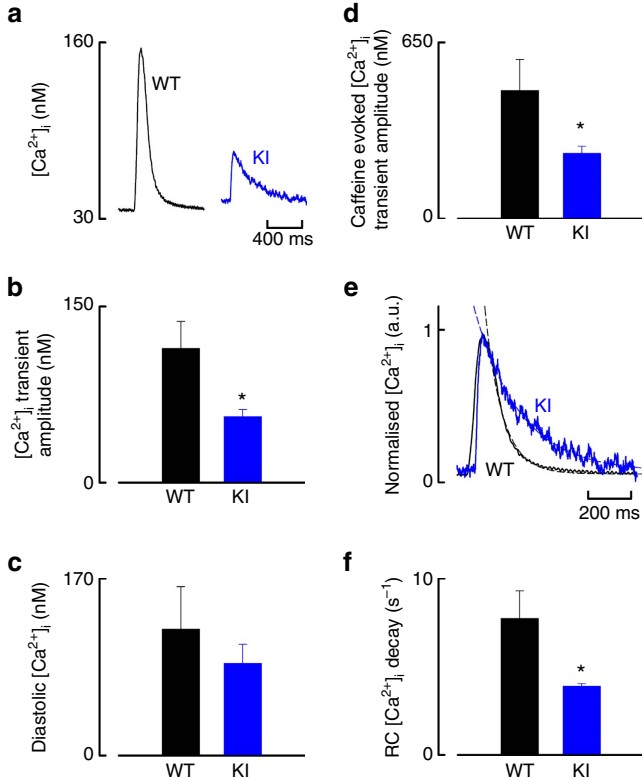

**Figure 3 | Comparison of intracellular calcium dynamics in WT versus C42S PKGIα KI mice.** (**a**) Specimen $[Ca^{2+}]_i$ transients from isolated ventricular cardiomyocytes. (**b,c**) Cells from KI mice showed significantly deficient mean systolic $[Ca^{2+}]_i$ transient amplitudes compared with WT (*$P < 0.05$; $n = 12$–13 from 4 to 6 hearts), whereas the mean diastolic $[Ca^{2+}]_i$ was not different between genotypes ($n = 16$ from 4 to 6 hearts). (**d**) Mean caffeine-evoked $[Ca^{2+}]_i$ transient amplitude (indicative of SR $Ca^{2+}$ content) was significantly reduced in cells from KI compared with WT (*$P < 0.05$; $n = 5$–13 from 3 hearts). (**e,f**) Normalised $[Ca^{2+}]_i$ transients permitting direct comparison of transient decay phase (indicative of SERCA2a activity) between genotypes. The dashed lines show single exponential fits, which were used to determine rate constant for the decay of $[Ca^{2+}]_i$ for each genotype, which is significantly slower in KI compared with WT (*$P < 0.05$; $n = 12$ cells from 4 to 6 hearts). Scale bars in **a** and **e** are 400 ms and 200 ms, respectively. Error bars show s.e.m. and $P$ values were determined by $t$-test.

pathways that diverge depending on the stimulus. To investigate PKGIα signalling in the heart, we monitored activation-dependent phosphorylation of PKGIα substrates using a chemical genetics phosphoproteomics approach[17,18]. The majority of proteins we identified, including the known PKGIα target heat shock protein beta 6 Ser16 (HspB6) (ref. 24), were reproducibly phosphorylated by cGMP-activated kinase. However, the crucial cardiac protein PLN was selectively and reproducibly phosphorylated at Ser16 by disulfide-activated PKGIα—not by cyclic nucleotide-activated kinase. This observation is in agreement with studies showing PKG can phosphorylate PLN (ref. 25), but this is not induced by stimuli that increase intracellular cGMP in cardiac muscle[26].

PKG isoforms are known to target binding partners via their N-terminal leucine zipper domains where the differing patterns of surface charge are important for regulating the interactions[27–29]. The Cys42 intermolecular disulfide of PKGIα is located within the leucine zipper domain and is likely to alter the physicochemical properties of this region. Depending on the substrate, formation of the Cys42 disulfide bond will promote or

inhibit kinase-substrate interactions, resulting in an increase or decrease in phosphorylation regardless of cGMP binding. Indeed, we found by ITC that disulfide-activation of PKGIα, in the absence of cyclic nucleotide, increased its affinity for the cytoplasmic domain of PLN at least five-fold.

Analysis of myocardium from WT or C42S PKGIα KI mice—which cannot form the activating disulfide bond—revealed a significant deficit in PLN Ser16 phosphorylation in the mutant tissue, consistent with our biochemical data. The reduced phosphorylation of PLN Ser16 produces substantial alterations in $Ca^{2+}$ handling. PLN binds to SERCA2a, the pump that mediates $Ca^{2+}$ reuptake into the SR, and reduces its activity. Therefore, SERCA2a activity influences the rate of decay of the $Ca^{2+}$ transient and the amount of $Ca^{2+}$ stored in the SR (SR $Ca^{2+}$ content). The latter controls the amplitude of the $Ca^{2+}$ transient, therefore SERCA2a activity also controls the amplitude of the $Ca^{2+}$ transient. PLN Ser16 phosphorylation relieves inhibition of SERCA2a resulting in increased activity, hastening the rate of decay of the $Ca^{2+}$ transient, increasing both the SR $Ca^{2+}$ content and the $Ca^{2+}$ transient amplitude. On the basis of these considerations, it is not surprising to find that the lower levels of PLN Ser16 phosphorylation observed in the C42S PKGIα KI are associated with a substantial reduction in the rate of decay of the $Ca^{2+}$ transient, the SR $Ca^{2+}$ content, as well as the $Ca^{2+}$ transient amplitude. PLN can be phosphorylated at Ser16 by cAMP-dependent protein kinase (PKA) following adrenergic stimulation. In addition, phosphorylation of Thr17 by CaMKII also increases SERCA2a activity[15]. The main physiological function of CaMKII-mediated phosphorylation is to adapt SERCA2a function to increases in heart rate. Identification of PLN Ser16 as a selective target of oxidized PKGIα raised a question about the functional significance and role of this phosphorylation event. We reasoned that oxidants produced during diastolic stretch (X-ROS signals)[13] may trigger disulfide-activation of PKGIα and subsequent phosphorylation of PLN Ser16. This stretch-dependent myocardial oxidant signalling should be deficient in the hearts of the KI mice, because of the inability of C42S PKGIα to transduce X − ROS signals into Ser16 PLN phosphorylation via the disulfide activation pathway. Indeed, we observed that cardiac stretch promoted oxidation of PKGIα in WT hearts, and was associated with translocation of the kinase to the SR fraction, and an increase in PLN Ser16 phosphorylation. These events were absent in the KI hearts, as hypothesized. On the basis of these observations, we speculated that the relationship between stretch and systolic contraction, that is, the Frank–Starling mechanism, might be impaired in the KI hearts. Indeed, contractile responses were markedly impaired in isolated KI hearts compared with WT as EDP (stretch) was progressively increased up to 10 mm Hg. The reduction in contractile responses is due both to impaired systolic function secondary to smaller $Ca^{2+}$ transients because of the lower SR $Ca^{2+}$ content, as well as diastolic dysfunction secondary to the slower $Ca^{2+}$ transient decay, which will impair cardiac filling. These observations clearly delineate a novel role for the oxidized PKGIα-PLN-SERCA2a axis in the modulation of the Frank–Starling mechanism, which traditionally has been attributed to modulation of myofilament properties.

At this point it is important to highlight an apparent discrepancy in our data. The observation that the systolic $Ca^{2+}$ transient was substantially lower in the KI cardiomyocytes compared with WT would be anticipated to manifest itself as attenuated cardiac systolic force development *in vivo* or in isolated heart preparations. However, this is not the case, as left ventricular systolic output measured by a PV catheter is identical between genotypes. This potential discrepancy can be fully reconciled by the fact that the KI is able to achieve the same

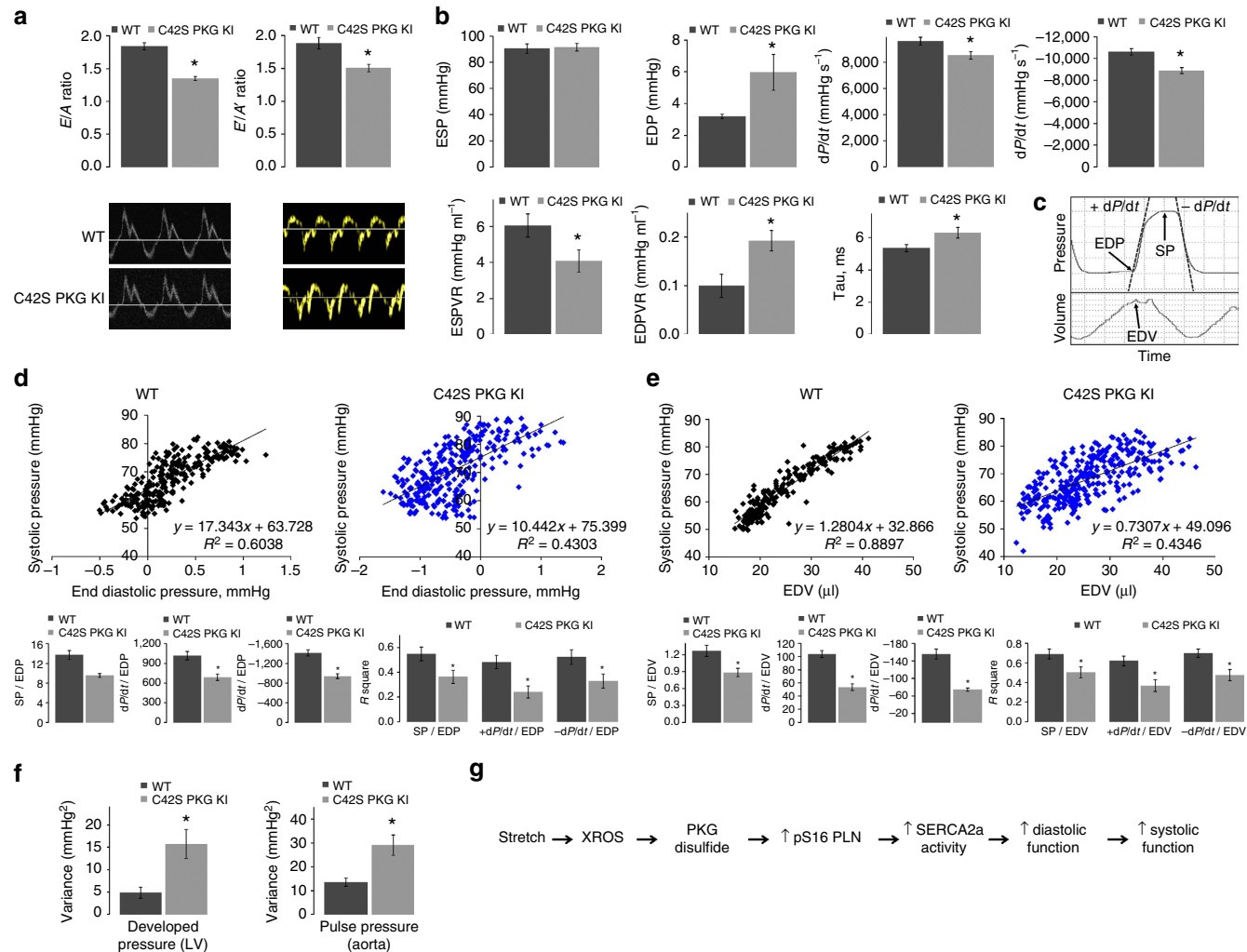

**Figure 4 | Impaired diastolic relaxation and Frank–Starling mechanism in C42S PKGIα KI mice *in vivo*. (a)** E/A and E′/A′ ratios for WT and KI mice measured by tissue Doppler echocardiography. Both ratios are significantly decreased in the mutant indicating diastolic dysfunction (*P < 0.05; n = 5). **(b)** Various cardiac parameters for the WT and KI mice derived from left ventricular PV loops. EDP, end-diastolic pressure-volume relationship and Tau were significantly increased in the KI, while the rates of relaxation and contraction and end-systolic pressure-volume relationship were significantly decreased (*P < 0.05; n = 9-10 for baseline measurements except inferior vena cava where n = 7-8). These observations are indicative of reduced contractility and impaired myocardial relaxation in the KI. **(c)** Representative trace of pressure and volume measured in the left ventricle of a WT or KI mouse over the time period of one cardiac cycle. EDP, SP, EDV, + dp/dt and − dp/dt were determined as indicated so that intra-beat relationships could be calculated that related directly to the Frank–Starling mechanism *in vivo*. **(d)** Representative scatter plot of SP versus EDP for a WT and KI mouse; data was obtained from 200 heartbeats as described above. Histograms show the averaged gradients for SP, + dp/dt and − dp/dt versus EDP and the corresponding mean coefficients of determination, $R^2$. Intra-beat relationships, as well as $R^2$ values were significantly decreased for each variable in the KI hearts (*P < 0.05; n = 9). **(e)** A representative scatter plot of SP versus EDV (an index of myocardial stretch) for a WT and KI mouse. Intra-beat relationships and associated $R^2$ values were determined as described above for EDP and, similarly, were significantly decreased for the KI in each case (*P < 0.05; n = 9). These results provide compelling evidence that the PKGIα Cys42 disulfide bond contributes to the Frank–Starling mechanism. **(f)** Variance in the LV developed pressure (SP − EDP; n = 9) and aortic pulse pressure (n = 7-10) in 1000 consecutive heartbeats for WT or KI mice. Both pressures were significantly more variable in the KI mice (*P < 0.05) further demonstrating dysregulation of cardiac output when PKGIα cannot be oxidant-activated. **(g)** Scheme showing how oxidation of PKGIα by stretch-induced oxidants contributes to the Frank–Starling response. Error bars show s.e.m. P values were determined by t-test.

systolic output as the WT *in vivo* at the cost of a sustained elevation in EDP. Similarly, isolated KI hearts can generate the same systolic pressure as the WT, but again this is only achieved by increasing their EDP above that required in WT. Thus at an EDP of 4 mm Hg KI hearts only generate ∼50% of the systolic output of WT, matching very well the proportional deficit in the systolic $Ca^{2+}$ amplitude measured in isolated unloaded cardiomyocytes. Essentially, the alterations to $Ca^{2+}$ cycling observed in the KI reflect the inability to engage the Frank–Starling mechanism as effectively as WT myocardium.

At very high EDP the preload will overcome the diastolic dysfunction and it is conceivable that additional mechanisms, not defined here, involving cardiac myofilaments or SR $Ca^{2+}$ load adaptation, participate to normalize systolic function.

We observed an increase in the pentamer/monomer ratio of PLN in the KI myocardium compared with the WT, which may represent an adaptive change in the KI. However, it is difficult to make firm conclusions about the significance of this observation, as there is evidence for refs 16,30, as well as against ref. 31, oligomerization mediating activation of SERCA2a. We should

also consider that there are other regulatory mechanisms that participate in the control of SERC2a activity, such as dephosphorylation of PLN by protein phosphatase 1, and by differential interactions with SR membrane proteins including sarcolipin and DWORF (refs 32–34).

An *in vivo* comparison of cardiac function in the WT and KI mice by echocardiography and ventricular PV loop analysis revealed diastolic dysfunction in the KI, consistent with the *ex vivo* heart preparations and intracellular $Ca^{2+}$ measurements. Taken together, our data supports a role for the PKGIα Cys42 disulfide bond in stretch-induced enhancement of myocardial relaxation to obtain the appropriate amount of ventricular filling during diastole (Fig. 4g). Analysis of high-resolution PV data from the WT and KI mice showed that as preload increased, the SP and rate of contraction for the next beat increased, as anticipated due to the Frank–Starling law of the heart. However, not only was the intra-beat relationship between preload and contractility diminished in the KI mice, systolic pressures and contraction rates were also more random for a given EDV. As a result, the pulse pressure was more variable in the KI. We concluded that oxidation of PKGIα is involved in coupling ventricular filling with cardiac output on a beat-to-beat basis, that is, it contributes to the Frank–Starling mechanism.

Deficiencies in the cardiac response to preload in the C42S PKGIα KI mice are due, at least in part, to insufficient phosphorylation of PLN Ser16 during ventricular filling. Removal of cytosolic $Ca^{2+}$ is therefore slower, which means that the myocardium cannot relax properly and a reduced SR $Ca^{2+}$ load also leads to diminished contractility[35]. Passive tension is also likely to play a role in the diminished Frank–Starling responses of the KI hearts due to increased interactions of the giant elastic protein titin with $Ca^{2+}$. $Ca^{2+}$ binding to titin is known to increase passive tension of the myocardium, making the ventricle stiffer and thus harder to fill[12]. Furthermore, because impaired relaxation results in inadequate extension of the sarcomeres, the myofilament $Ca^{2+}$ sensitivity—which is dependent on sarcomere length[11,12,29]—will be reduced in the KI cardiomyocytes and also contribute to the decreased contractility observed in the KI hearts[11,12,36].

Other molecular mechanisms that facilitate Frank–Starling responses may also be altered in the absence of the PKGIα oxidation pathway. For example, phosphorylation of the sarcomeric proteins cTnI and cMyBP-C is involved in length-dependent activation of myofilaments[37]. However, we did not observe changes in phosphorylation of key residues in these proteins in the KI hearts. Basal phosphorylation of several other phosphosites involved in cardiac EC coupling and $Ca^{2+}$ handling including phospholemman Ser68 and CaMK2 Thr282 was also unchanged in the KI hearts, suggesting that these residues are not central to the X-ROS/PKGIα oxidation/enhanced myocardial relaxation pathway. S-nitrosylation of PLN is another modification that has been shown to modulate the Frank–Starling mechanism and it is possible that this modification has a role in X-ROS signalling[38]. However, we cannot envisage how the PKGIα Cys42Ser mutation could affect PLN S-nitrosylation and it is unlikely that this redox modification contributes to the diminished Frank–Starling mechanism observed in the KI mice.

In conclusion, fundamental to the Frank–Starling law of the heart is an initiating diastolic stretch which induces events that result in a systolic contraction of appropriate force. Here, as summarized in Fig. 4g, we show that this crucial relaxation step is significantly mediated by oxidative activation of PKGIα which phosphorylates phospholamban to enhance diastolic relaxation. Furthermore, in the absence of this redox control mechanism, as is the case in the KI, the pressure amplitude the heart generates from beat-to-beat is erratic.

## Methods

**Animal studies.** All procedures were performed in accordance with the United Kingdom Home Office Guidance on the Operation of the Animals (Scientific Procedures) Act 1986. The KI mice constitutively expressing PKGIα C42S were generated on a pure C57BL/6 background by TaconicArtemis (Germany) as described previously[5]. All mice used in this study were male and age and body weight–matched.

**Method for identification of direct substrates of PKGIα.** We employed a chemical genetic method[18] to identify direct substrates of cGMP-activated PKGIα and disulfide bond-activated PKGIα in heart tissue. This method involves mutation of the ATP binding-site of the kinase of interest so that it can accept a 'bulky' N6-alkylated ATP analogue for example, N6$^−$phenylethyl ATP. An oxygen atom on the γ phosphate is also replaced with a sulfur atom—giving an N6-alkylated ATPγS analogue—so that the mutant kinase catalyses the transfer of a thiophosphate group ($–PO_3S^{3−}$) to its substrates instead of a phosphate. The thiophosphate group is nucleophilic, providing a basis for substrates of the analogue-sensitive kinase to be purified by a 'covalent capture' protocol. LC–MS/MS allows identification of the substrate and localization of the phosphorylation site. Thiophosphorylated proteins can also be detected with a thiophosphate ester-specific antibody[39]. Analogue-sensitive mutants of PKGIα and PKGIα C42S (which cannot form the activating intermolecular disulfide bond) were generated by mutating Met438 in the ATP-binding domain to Gly. This mutation was chosen based on a previous study where an analogue-sensitive mutant of the catalytic subunit of PKA—which shares sequence homology with PKGIα—was engineered[40]. Detailed protocols are given below.

**Recombinant WT and analogue-sensitive PKGIα mutants.** A pCDNA3 expression vector encoding human FLAG-tagged PKGIα (ref. 41) underwent site-directed mutagenesis to generate constructs for untagged WT, C42S, M438G and C42S/M438G PKGIα. Mutations were introduced using the QuikChange II Site-Directed Mutagenesis Kit (Agilent) according to the manufacturer's instructions. Expression and purification of the PKGIα mutants was performed according to a published method[42] as follows: suspension FreeStyle 293-F cells (ThermoFisher Scientific) were transfected with the appropriate PKGIα construct using the stable cationic polymer polyethyleneimine (PEI) as a transfection reagent[43]. After ~72 h cells were harvested by centrifugation (400g; room temperature; 15 min), re-suspended in ice cold lysis buffer (25 mM sodium phosphate buffer pH 6.8; 10 mM ethylenediaminetetraacetic acid (EDTA); 100 mM NaCl; 10 mM benzamidine hydrochloride; and 10 mM dithiothreitol (DTT)) and frozen in liquid $N_2$. Cells were lysed by three freeze (liquid $N_2$)-thaw (37 °C) cycles and the lysate was clarified by centrifugation at 140,000g and 4 °C for 30 min. The soluble protein fraction was loaded onto a pre-equilibrated 8-(2-aminoethylamino)adenosine-3′, 5′-cyclic monophosphate column (8-AEA-cAMP agarose; BioLog, Germany) followed by washing with 20 column volumes of lysis buffer. The column was further washed with lysis buffer $+ 3$ M NaCl (5 column volumes) and PKGIα was then eluted with lysis buffer $+ 150$ mM NaCl and 500 μM cAMP. Removal of cAMP and buffer exchange was achieved by extensive dialysis against 25 mM sodium phosphate buffer pH 6.8, 2 mM EDTA and 100 mM NaCl. For ITC analysis, WT PKGIα was incubated with 10 mM lipoic acid for 2 h on ice to obtain ~100% disulfide dimer (confirmed by SDS-PAGE). Oxidized WT or C42S PKGIα was further purified by size exclusion chromatography using a HiLoad 16/600 Superdex 200 pg column (GE Healthcare) with 50 mM sodium phosphate buffer pH 7.4 and 100 mM NaCl. Proteins were concentrated using Amicon Ultra centrifugal filter devices (Merck Millipore). DTT was absent in storage buffers so that the activating disulfide bond would not be reduced and so that the PKGIα M438G mutant would be oxidized to disulfide dimer in the presence of air (confirmed by SDS-PAGE). Protein concentration was determined by Pierce BCA assay (ThermoFisher) and enzyme activity was confirmed using the Omnia kinase assay kit (ThermoFisher).

**Thiophosphorylation of PKGIα Substrates in Heart Homogenate.** Male Wistar rats (9–10 weeks; body weight 300–330 g) were euthanized by intraperitoneal injection of sodium pentobarbitone (200 mg kg$^{−1}$) with heparin (500 USP units). Hearts were flushed in the chest with ice cold Krebs buffer and the left ventricle was excised and immediately transferred to ice cold homogenization buffer (2 ml 1$^{−1}$ g; 50 mM sodium phosphate buffer pH 7.4, 150 mM NaCl, 0.1% Tween 20 and EDTA-free protease inhibitor cocktail tablet (Roche)). Tissue was homogenized with a Ystral homogenizer and the homogenate was clarified by centrifugation at 50,000g and 4 °C for 30 min. The protein concentration of the soluble fraction was determined by BCA assay and then adjusted to 20 mg ml. Four thiophosphorylation reactions were set up: (1) with cGMP-activated PKGIα C42S/M438G, (2) with disulfide-activated PKGIα M438G, (3) with unactivated, basal, PKGIα C42S/M438G and (4) a 'no kinase' control. The total reaction volume was 200 μl and the mixtures consisted of 25 mM Tris pH 7.5; 10 mM $MgCl_2$; ± 100 μM cGMP (added to the cGMP-activated PKGIα reaction only); 0.4 mM ATP; 6 mM GTP; 1 mM N6-furfuryladenosine-5′-O-(3-thiotriphosphate) (6-Fu-ATP-γ-S; BioLog, Germany); 100 μl of heart homogenate (soluble fraction) and 20 μg of the appropriate analogue-sensitive PKGIα (not added to the

control reaction). The kinase reactions proceeded for 1 h at 30 °C, after that time they were quenched by addition of 220 mM EDTA. 10 μl of each mixture was taken for alkylation with 10 mM p-nitrobenzyl mesylate (PNBM; Abcam) to check the reaction by Western blot using the anti-thiophosphate ester antibody (51-8) (ab92570; Abcam; working concentration of 1:5,000 and secondary antibody used at 1:10,000). The protocol was repeated four times; each time with a fresh rat heart (that is, four biological replicates; 16 kinase reactions in total). Samples were frozen in liquid $N_2$ and stored at $-80$ °C until the 'covalent capture' procedure.

**Covalent capture of thiophosphorylated peptides.** Thiophosphorylated peptides were isolated and converted to phosphopeptides for analysis by LC–MS/MS according to a published method[18] as follows: proteins were denatured by the addition of 120 mg solid urea (60% w/v) and 10 mM tris(2-carboxyethyl) phosphine (TCEP), with incubation at 55 °C for 1 h. Samples were diluted 2× with 50 mM ammonium bicarbonate and additional TCEP was added to a final concentration of 10 mM. The pH was adjusted to 8, and trypsin (sequencing grade; Promega) was added at a substrate to protease ratio of 20:1. Proteins were digested for ∼16 h at 37 °C, after that time the trypsin was quenched by acidification of the samples to ∼pH 3 with trifluoroacetic acid (TFA). Resulting thiophosphopeptides were desalted with Sep-Pak $C_{18}$ columns (Waters); peptides were eluted with 70% acetonitrile and 0.1% TFA (1 ml) and concentrated to ∼40 μl in a SpeedVac concentrator. For the covalent capture step, the peptides were diluted to ∼200 μl with 200 mM HEPES pH 7 and acetonitrile to a final concentration of ∼50 mM and 50%, respectively. The pH was adjusted to 7, and samples were subsequently loaded onto 100 μl of SulfoLink coupling resin (agarose activated with iodoacetyl groups; Thermo Scientific) pre-equilibrated with 200 mM HEPES pH 7 and 25 μg of BSA (to reduce nonspecific binding). The coupling reaction was left in the dark for ∼16 h at room temperature with end/end rotation. The beads were then collected in 1 ml fritted-tubes (Sigma-Aldrich) and washed with 1 ml each of water, 5 M NaCl, 50% acetonitrile, 5% formic acid and 10 mM DTT. Phosphopeptides were eluted from the resin with 200 μl of 1 mg ml$^{-1}$ oxone solution (∼pH 4; Sigma-Aldrich), which was allowed to rest on the beads for 30 min, followed by a further 50 μl of oxone solution. Samples were desalted using $C_{18}$ ZipTip pipette tips (Merck Millipore); phosphopeptides were eluted with 20 μl X3 of 50% acetonitrile and 0.1% TFA and concentrated to ∼10 μl in a SpeedVac concentrator, ready for injection on to the LC column.

**LC–MS/MS and data analysis.** The abundance of phosphopeptides in each of the kinase reaction mixtures described above was determined by a label-free quantitative phosphoproteomic analysis[19,20]. LC–MS/MS analysis was performed on an Ultimate 3000 nLC system (ThermoFisher) connected to an LTQ Orbitrap XL instrument. Samples were injected onto an Acclaim PepMap100 $C_{18}$ pre-column (5 μm, 100 Å, 300 μm i.d. × 5 mm) and washed for 3 min with 90% buffer A ($H_2O$ and 0.1% (v/v) formic acid) at a flow rate of 25 μl min$^{-1}$. Reversed-phase chromatographic separation was performed on an Acclaim PepMap100 $C_{18}$ Nano LC column (3 μm, 100 Å, 75 μm i.d. × 25 cm) with a linear gradient of 10–50% buffer B (ACN and 0.1% (v/v) formic acid) for 90 min at a flow rate of 300 nl per min. Survey full scan MS spectra (from m/z 390-1700) were acquired in the Orbitrap with a resolution of 60,000 at m/z 400. The mass spectrometer was operated in the data-dependent mode selecting the six most intense ions for CID. For phosphopeptide analysis, multi-stage activation for neutral loss of masses 97.97, 48.985 and 32.65667 was enabled. Target ions selected for MS/MS were dynamically excluded for 15 s. For accurate mass measurement, the lock mass option was enabled using the polydimethylcyclosiloxane ion (m/z 455.12003) as an internal calibrant. MS/MS spectra were de-isotoped and peak lists generated with Mascot Distiller (v2) and searched against the SwissProt database (2014_06; rat; 7,917 entries) using Mascot server (v2.4.1). Quantification was performed from the extracted ion chromatograms using Pescal software[44,45]. Allowed time and mass windows for the extracted ion chromatogramss were 1.5 min and 7 p.p.m. respectively. Substrates that were thiophosphorylated by endogenous kinases present in the heart homogenate—rather than the analogue-sensitive PKGIα mutants—were manually identified by comparison of the 'no kinase' normalized peak areas (that is, abundances) with phosphopeptide abundances in the corresponding cGMP, disulfide or unactivated-PKGIα samples, and omitted from further data analysis. Phosphopeptide abundances were then compared between basal, cGMP-activated and disulfide-activated PKGIα groups. One-way ANOVA was performed for each phosphopeptide followed by a post-hoc Dunnett's test to indicate statistically significant differences between the basal and activated mutant kinase groups.

**Langendorff perfusion of isolated mouse hearts.** Mice were euthanized by intraperitoneal injection of 6.6% sodium pentobarbitone (250 mg kg$^{-1}$) pre-mixed with heparin (500 USP units). Hearts were rapidly excised, immediately mounted onto Langendorff apparatus, and retrograde perfusion was established at a constant pressure of 80 mm Hg with Krebs-Henseleit buffer (in mM: 118.5 NaCl, 25.0 NaHCO₃, 4.75 KCl, 1.18 KH₂PO₄, 1.27 MgSO₄, 11.0 D-glucose and 1.4 CaCl₂) equilibrated with 95% $O_2$ and 5% $CO_2$ at 37 °C. Hearts were paced at 550 b.p.m. A fluid-filled balloon inserted into the left ventricle was used to monitor contractile

function. Hearts were stabilized for 20 min before stepwise inflation of the balloon to give increments in EDP which was measured via a pressure transducer. In some experiments hearts were stabilized with a deflated balloon, which was then left deflated or was inflated to 5 mm Hg for another 10 min to measure the effect of stretching. At the end of the protocol hearts were rapidly dismounted and frozen in liquid nitrogen until further analysis.

**Fractionation and Immunoblotting.** Hearts were homogenized in ice cold Tris–HCl pH 7.4, 100 mM maleimide (included to alkylated thiols and 'freeze' protein oxidation state) and EDTA-free protease inhibitor tablet (Roche) with a Ystral homogenizer. Heart homogenate was separated into soluble and particulate fractions by 15 min centrifugation at 25,000g. Reducing agent was not included in the SDS-PAGE sample buffer when the oxidation state of PKG1α was to be analysed by Western blot. Unless stated otherwise, samples intended for PLN immunoblots were boiled for at least 5 min immediately before loading the gel. The working concentration of all antibodies used in this study was 1:1000 unless stated otherwise. Primary antibodies were for PKG (ADI-KAP-PK005; Enzo), PLN (A010-14; Badrilla; 1:10,000), PLN pSer16 (A010-12; Badrilla; 1:5000), PLN pThr17 (A010-13AP; Badrilla; 1:5000) cTnI (4002; Cell Signaling Technology), cTnI Ser22/23 (4004; Cell Signaling Technology), cMyBP-C (sc-137180; Santa Cruz), cMyBP-C Ser282 (ALX-215-057-R050; Enzo), RyR2 Ser2808 (A010-30AP; Badrilla), phospholemman (custom-made FXYD1, FXYD1 pSer63, FXYD1 pSer68 and FXYD1 pSer69)[46], MLC2 (3672; Cell Signaling Technology), MLC2 Ser19 (3675; Cell Signaling Technology), CaMK2-β/γ/δ (SAB4503244; Sigma), CaMK2-β/γ/δ Thr282 (SAB4504607; Sigma), heavy chain cardiac myosin (ab50967; Abcam) and slow myosin heavy chain (MAB1628; Sigma). Horseradish peroxidase–linked rabbit or mouse secondary antibodies (Cell Signaling Technology) and ECL Western Blotting Detection Reagent (GE Healthcare) were used. Digitized immunoblots were analysed with Gel-Pro Analyzer 3.1 software. Uncropped blots are displayed in Supplementary Figs 1 and 2. The percentage of PKGIα disulfide dimer was quantified from total PKG1α protein expression and phosphorylation was normalized to total protein levels where possible.

**Isothermal titration calorimetry.** ITC experiments were carried out on a high sensitivity MicroCal iTC200 microcalorimeter (Malvern Instruments, UK). A synthetic N-terminally acetylated peptide corresponding to the N-terminal cyto-plasmic domain of human phospholamban, PLN (amino acid 1-23) was purchased from PeptideSynthetics (purity >95%; Peptide Protein Research Ltd, UK). Titra-tions were performed at 25 °C in 50 mM sodium phosphate buffer pH 7.4 and 100 mM NaCl. Aliquots of 22 μl PLN (1–23) (700 μM) were titrated into the reaction cell containing WT disulfide PKGIα or the C42S mutant (70 μM) at 150 or 180 s intervals. Integrated heat data was fitted to a theoretical titration curve using a nonlinear least-squares minimization algorithm in the MicroCal-Origin 7.0 software package as previously described.

**Isolated cardiomyocyte Ca$^{2+}$ measurements.** Ventricular myocytes were iso-lated from 3 to 4-month-old C42S PKGIα KI mice and their WT littermates using an established enzymatic digestion technique[47] as follows: mice were killed by cervical dislocation and hearts excised and placed in ice-cold isolation solution containing (in mM) NaCl, 134; HEPES, 10; Glucose, 11.1; NaH₂PO₄, 1.2; MgSO₄, 1.2 and KCl, 4; at pH 7.34. The aorta was cannulated and the heart was retrogradely perfused on a Langendorff apparatus with calcium free isolation solution for 5 min at 37 °C. Collagenase (1 mg ml$^{-1}$, type II, Worthington Biochemical Cooperation, NJ, USA) was added to the perfusate for 7–10 min. Following the digestion, ventricles were removed and minced in Taurine solution containing (in mM) NaCl, 115; HEPES, 10; Glucose, 11.1; NaH₂PO₄, 1.2; MgSO₄, 1.2; KCl, 4 and Taurine, 50; at pH 7.34 and then filtered through a 200 μm pore size mesh. Gentle agitation was also used to release single myocytes. CaCl₂ was then gradually added to restore resting Ca levels. Cells were stored in an experimental solution and kept at room temperature before use.

All experiments were performed at 37 °C. Cells were electrically stimulated at 1 Hz by field stimulation. The superfusion solution contained (in mM) NaCl 135, glucose 11, CaCl₂ 1, HEPES 10, MgCl₂ 1, KCl 4, probenecid 2; titrated to pH 7.4 with 2 mol l$^{-1}$ NaOH. The probenecid was required to reduce loss of fluorescent indicators from the cell, a particular problem at 37 °C.

To measure cytosolic Ca$^{2+}$ levels, cells were loaded with the acetoxymethyl (AM) ester form Fluo-3 (Molecular Probes) and excited continuously at 488 nm. Emitted fluorescence was measured with a 515 nm long-pass filter. Raw fluorescent signals were calibrated off-line as per equation (1). At 37 °C, the $K_d$ of Fluo-3 was taken to be 864 nm. Background fluorescence was subtracted from all raw signals. Custom-written Excel routines were used to measure Ca$^{2+}$ transient amplitude, diastolic Ca$^{2+}$ levels, rate of decay of the Ca$^{2+}$ transient and SR Ca$^{2+}$ content[48]. SR Ca$^{2+}$ content was estimated from the amplitude of the Ca$^{2+}$ transient evoked by application of 10 mM caffeine.

$$[Ca] = \frac{F \times K_d}{F_{max} - F} \quad (1)$$

**Echocardiography.** Mice were anesthetized and exmined by echocardiography using a high-resolution Vevo 770 echocardiography system (VisualSonics) with a RMV-707B transducer running at 30 MHz. High-resolution images were obtained for offline measurements with Vevo Software (VisualSonics). For the assessment of diastolic function, an apical four-chamber view was acquired by positioning the transducer as parallel to the mitral inflow as possible. Tissue motion velocity was assessed by spectral pulsed-wave Tissue Doppler imaging, obtained from the mitral septal annulus in the parasternal short axis view. LV diastolic function was assessed by the measurement of the LV transmitral early peak flow velocity ($E$) to LV transmitral late peak flow velocity ($A$) wave ratio and mitral annulus early diastole tissue motion ($E'$) to mitral annulus late diastole tissue motion ($A'$) wave ratio.

**In vivo pressure-volume analysis and blood pressure analysis.** Invasive pressure-volume analysis real-time pressure volume loops were obtained using the ADVantage system (Scisense Inc., Canada) which uses a miniaturized 1.2 Fr admittance catheter. In our experiments measurements of LV function were performed in the more physiological closed chest mode when a catheter was placed in LV by retrograde approach. Briefly, mice were anesthetized, right internal carotid artery was exposed and catheterized. The 1.2 Fr catheter was advanced towards the heart and inserted into the LV cavity via the aortic valve. In order to analyse the effect of preload changes, inferior vena cava occlusion was performed. Blood pressure and heart rate were assessed by telemetry in conscious mice. Mice were anesthetized with isoflurane and a TA11PA-C10 probe catheter (Data Science International) was implanted into the aortic arch via the left carotid artery. Blood pressure variability was analysed from a continuous telemetric blood pressure record made in undisturbed telemetered animals in a quiet room. Thousand beats were chosen for analysis in each mouse.

**Statistics.** All results relating to the WT and KI mice are presented as mean ± s.e. of the mean (s.e.m.). Differences between groups were assessed by ANOVA followed by post-hoc $t$-test. Differences were considered significant at the 95% confidence level. In PV loop analysis measurements > 2 s.d.s from the mean were excluded, which resulted in one mouse per group being omitted.

**Data availability.** The data that support the findings of this study are available from the corresponding author on request.

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

## Acknowledgements

This work was supported by the British Heart Foundation, European Research Council (ERC Advanced award), Medical Research Council and the Department of Health via the NIHR cBRC award to Guy's & St Thomas' NHS Foundation Trust. Part of the work was also supported by the NIHR University College London Hospitals Biomedical Research Centre. ITC experiments were carried out at the Centre for Biomolecular Spectroscopy, King's College London, established with a Capital Award from the Wellcome Trust. Kornel Kistamas was supported by the BHF Chair in Cardiac Physiology to Professor David Eisner.

## Author contributions

J.S. and O.P. contributed equally to this study. J.S. designed and performed the majority of the *in vitro* kinase experiments, as well as the work up of samples for analysis by mass spectrometry, with assistance from E.D.M. M.R.C. performed, analysed and interpreted ITC studies with assistance from J.S. who also prepared the recombinant kinase. J.W., J.S., P.R.C. or J.F.T. designed, performed or interpreted data from the phosphoproteomics mass spectrometry analysis. O.P. designed and performed the majority of the biochemical analyses of cardiac tissue with assistance from F.C. O.P. also performed, analysed and interpreted the isolated heart and echocardiography studies. O.R. and O.P. performed the blood pressure analysis. A.B. performed the pressure-volume analysis and analysed and interpreted the data with O.P. M.J.S. and M.S.M. analysed and interpreted data from the *in vivo* and *ex vivo* cardiac function studies. K.K., N.H., D.J.G, A.G. and L.V. designed, performed or interpreted data from isolated myocyte calcium measurements. P.E. conceived and coordinated the study, designed experiments, analysed and interpreted the data. P.E. also wrote the paper with J.S. and O.P., with input or editing from all authors.

## Additional information

**Competing financial interests:** The authors declare no competing financial interests.

