## [Peer Review File · Nature Communications]

Reviewers' comments:

Reviewer #1 (expert in heart function and physiology)

Comments to the Author

This is an interesting manuscript that identifies Phospholamban Ser16 as a selective target of disulfide-activated PKG. The authors show that hearts unable to form the activating disulfide bond (C42S PKGI α mice) display an impaired Frank-Starling responses. Indeed, the authors report a decrease in the relaxation and contraction rate responses to EDP increases in the C42S PKGI α hearts. They also show an impaired myocardial function in the knock-in mice in vivo. The study is overall well performed, well presented and the results correctly interpreted.

Following are specific comments that need to be addressed

- Figure 1a: the phosphorylation of several proteins is decreased when PKG is activated by Cys42 disulfide bond. Is this decrease significant for some of the proteins? If yes, these proteins should be reported in Table1 and the authors should discuss how PKG activation could decrease the phosphorylation of these proteins.
- Figure 1a: what about the phosphopeptides that seem to be increased upon PKG activation by disulfide bond and unchanged upon PKG activation by cGMP? Please clarify.
- Table1: fold change values should be reported.
- Figure 2: the authors show that an increase in EDP induces an increase in PLN phosphorylation. It would be important to also determine Serca activity in these conditions.
- The authors should determine the SR Calcium levels in WT and KI mice.
- In order to confirm that the oxidation of PKG is mediating Ser16PLN phosphorylation, the effect of H₂O₂ on Ser16PLN phosphorylation and Serca activity in WT and C42S PKGI α isolated cells should be shown.

Reviewer #2 (expert in redox chemistry and cardiovascular disease)

Comments to the Author

Comments for Author

--The authors should determine the phospholamban K_d for the disulfide form of PKG and for the C42S PKG mutant.

--What is the reversibility of the disulfide oxidation of cys42 in PKG?

--What are the consequences of cys42 oxidation for cGMP activation of PKG?

--The authors should perform formal P-V loop analysis of the LV for the WT and C42S PKG mutant.

--The authors do not consider other potential molecular mechanisms that may in part account for the Frank-Starling effect, including phospholamban S-nitrosylation (cf. Garofalo F et al., Proc Biol Sci 2009;276:4043-52-in fish) or cardiac myosin-binding protein C and TnI phosphorylation (Kumar M et al., JBC 2015;290:29241-9), to name but two. How does PKG cys42 oxidation compare to these

potential regulatory mechanisms in governing the overall phenomenon?

Reviewer #3 (expert in phospholamban)

Comments to the Author

The paper by Scotcher et al report a chemical screen showing that PKG targets phospholamban, that PKG KI mice have reduced Frank Starling responses, and conclude that the impaired response is due to a phosphorylation of PLN.

There are aspects of this study that have merit and are novel. But, as presented, this manuscript is not acceptable for publication.

The data in the PKG KI mice by isolated in vitro hemodynamics is clear (Fig 2a,b,d,e).

Major concerns

1. The potential role of PKG in regulating PLN to contribute to Frank Starling responses would be interesting. However, challenging the conventional idea that the response is primarily due to tension-length aspects, rather it is due to direct phosphorylation status MUST be proven. The authors made no effort to document myosin length measures, nor phosphorylation status of myosin! This would be a critical fatal flaw that would need to be addressed.

2. For control studies in the PKG KI mouse, the authors must provide additional controls, not just examine the status of PLN. I fully appreciate they identified one peptide for PLN-pS and that is their focus. However, RyR, DHPR, CamKII, myosin heavy chain, as examples, MUST be examined with respect to their total levels and their phosphorylation status. This would be a basic fundamental requirement.

3. The final figure 3g has absolutely no supporting data in their paper. In fact, the authors concluded "Scheme showing how oxidation of PKGI α by stretch induced oxidants mediates the Frank-Starling response" This paper presents no direct evidence that ROS levels were elevated in their model, nor that SERCA activity was increased. To make these conclusions, the authors would need to measure ROS levels in their systems, perhaps scavenge ROS levels, assess Ca²⁺ store levels, assess Ca²⁺ transients, examine responses with/without ISO treatment.

Minor concerns

-Formatting of this manuscript made it extremely difficult to read. For instance, there is no introduction. Rather the paper goes from a general abstract directly into an Intro/Results/Discussion. Figure legends are not conventional, rather an expanded results section. Guidelines for NComm submission are online and very clear. As a result, the authors had an opportunity to frame their discovery in perspective but failed to do so.

-Echo data in Figure 3 a/b might be better presented as a standard table, as in most studies.

-Fig 1b. were these blots boiled, what ratio of pentamer/monomer was documented?

We would like to thank the editor and the reviewers for their logical as well as valuable comments and feedback. In this revision we have included significant additional data from new experiments as suggested by the reviewers. Some of these relate to isolated cardiomyocyte calcium measurements, which required us to work with a collaborator lab in Manchester who had these methods up-and-running. The transgenic colony had to be established there, and this explains why we were delayed in providing this revision after what we considered a positive first review. We think the revised manuscript is substantially improved as a result of addressing the reviewers' points. We provide detailed responses to each reviewer comment below, explaining what we have done in light of their feedback.

Reviewer #1 (expert in heart function and physiology)

Comments to the Author

This is an interesting manuscript that identifies Phospholamban Ser16 as a selective target of disulfide-activated PKG. The authors show that hearts unable to form the activating disulfide bond (C42S PKG1 α mice) display an impaired Frank-Starling responses. Indeed, the authors report a decrease in the relaxation and contraction rate responses to EDP increases in the C42S PKG1 α hearts. They also show an impaired myocardial function in the knock-in mice in vivo. The study is overall well performed, well presented and the results correctly interpreted.

Following are specific comments that need to be addressed:

- Figure 1a: the phosphorylation of several proteins is decreased when PKG is activated by Cys42 disulfide bond. Is this decrease significant for some of the proteins? If yes, these proteins should be reported in Table 1 and the authors should discuss how PKG activation could decrease the phosphorylation of these proteins.

Disulfide-dependent decreases in phosphorylation were not statistically significant for any of the PKG1 α substrates that we identified. All proteins that had a statistically significant change in phosphorylation upon PKG1 α activation are listed in Table 1—the text in the results section has been amended to clarify this:

“Substrate phosphorylation was assessed when PKG1 α was activated by the Cys42 disulfide bond, or via the classical pathway with cGMP, and compared to phosphorylation when PKG1 α was in its basal “unactivated” state (control). 85 direct substrates of PKG1 α were identified (Figure 1a and Supplementary Information) and the 29 substrates that had a statistically significant change in phosphorylation upon activation of the kinase are listed in Table 1. Phosphorylation of 28 of these proteins was significantly increased when PKG1 α was activated by cGMP. Intriguingly, the abundance of a phosphopeptide from the cardiac protein phospholamban (PLN pSer16; RApSTIEMPQQAR) was significantly increased relative to control when PKG1 α was activated by Cys42 oxidation rather than by cyclic nucleotide binding, indicating that PLN Ser16 is a selective target of disulfide dimer PKG1 α ”.

Table 1 has also been amended and now lists the fold changes for each substrate and indicates which fold changes were significant.

The scatter plot in Figure 1a also indicates statistically significant changes in phosphorylation by colour coding (pink circles for cGMP-induced increases and blue for disulfide-induced increases) and we have amended the Figure legend to explicitly state that:

“Decreases in phosphorylation were not statistically significant for any substrate.”

Even though decreases in phosphorylation were not statistically significant, it is indeed plausible that disulfide activation of PKGI α could inhibit phosphorylation of certain substrates and we have included a brief explanation about how this might occur in the Discussion:

“PKG isoforms are known to target binding partners via their N-terminal leucine zipper domains where the differing patterns of surface charge are important for regulating the interactions^{27, 28, 29}. The Cys42 intermolecular disulfide of PKGI α is located within the leucine zipper domain and is likely to alter the physicochemical properties of this region. Depending on the substrate, formation of the Cys42 disulfide bond will promote or inhibit kinase-substrate interactions, resulting in an increase or decrease in phosphorylation regardless of cGMP binding. Indeed, we found by ITC that disulfide-activation of PKGI α , in the absence of cyclic nucleotide, increased its affinity for the cytoplasmic domain of PLN at least 5-fold.”

- Figure 1a: what about the phosphopeptides that seem to be increased upon PKG activation by disulfide bond and unchanged upon PKG activation by cGMP? Please clarify.

There were ~3 phosphopeptides whose abundances were increased when PKGI α was activated by disulfide bond but unchanged when PKGI α was activated by cGMP (black circles above x axis and on/very close to the y axis; LYR motif containing protein 1 Ser110, rho GTPase-activating protein 35 Ser1150, and junctophilin-2 Ser240). However, the increased abundances were not statistically significant therefore these proteins are not listed in Table 1—only phosphopeptides that had a reproducible change in abundance ($p < 0.05$) are stated in the main results section.

As detailed in the reply to the previous comment, we have altered text in the Results section, the caption for Figure 1, and expanded Table 1, to clarify precisely what phosphopeptides had a statistically significant fold change associated with them and whether this fold change was positive or negative for cGMP or disulfide activation.

We have added some content into the Discussion to explain how a protein could potentially be phosphorylated by disulfide PKGI α but not by cGMP-activated PKGI α (see reply to previous comment).

- Table1: fold change values should be reported.

Amended. Log₂ fold changes are also given for all 85 identified substrates in the Supplementary Information excel file where the complete proteomics dataset is listed.

- Figure 2: the authors show that an increase in EDP induces an increase in PLN phosphorylation. It would be important to also determine Serca activity in these conditions. The authors should determine the SR Calcium levels in WT and KI mice. In order to confirm that the oxidation of PKG is mediating Ser16PLN phosphorylation, the effect of H₂O₂ on Ser16PLN phosphorylation and Serca activity in WT and C42S PKGI α isolated cells should be shown.

Unfortunately, our attempts to establish a biochemical SERCA2a activity assay using mouse heart failed. Pilot studies showed that we would have to pool 6-7 mouse hearts to get sufficient tissue to perform a single assay. With perhaps 6 replicates per group and two genotype each with or without stretch this would mean $7 \times 6 \times 2 \times 2 = 164$ mice would be needed to perform this analysis, which was not possible due to limited resources. There are also ethical considerations related to the use of such large numbers of animals to address one point that can be significantly addressed by other means. For example, the changes in phosphorylation of PLN at serine 16 is commonly regarded as a measure of SERCA2a activity, which when put together with all the other observations reported here, provide significant support for the conclusion that SERCA2a is less active in the KI. However, importantly, in addition to this cumulative evidence that supports SERCA2a being less active in KI, this point is directly addressed by the isolated cardiac myocytes studies requested by the reviewer and commented on in more detail below. In mouse cardiac myocytes the rate of calcium decay from the cytosol during diastole is principally mediated by SR uptake as a result of SERCA2a activity. In this connection the calcium decay rate was ~50% slower in KI compared to WT (see Figure 3 in the revised manuscript). The isolated myocyte studies also allowed us to compare the SR content of WT versus KI cardiac myocytes, and this showed the calcium content was also ~50% lower in KI compared to WT. This is consistent with the attenuated rate of uptake of calcium in the KI. A significant amount of additional text has been added to the Results and Discussion sections to address this important point:

“Ca²⁺ handling is altered in isolated cardiomyocytes from C42S PKG1 α KI hearts

*Experiments were performed in ventricular myocytes isolated from adult WT or C42S PKG1 α KI hearts, comparing intracellular calcium ($[Ca^{2+}]_i$) dynamics between genotypes. Specimen transients (**Figure 3a**) are clearly consistent with significantly altered Ca²⁺ handling in the cells from KI animals. Quantitative analysis of the transients showed the KI was significantly deficient in their systolic $[Ca^{2+}]_i$ transient and SR Ca²⁺ content evoked by application of caffeine, whereas their diastolic $[Ca^{2+}]_i$ levels were the same between genotypes (**Figure 3b-d**). Normalisation of the $[Ca^{2+}]_i$ transients allowed direct comparison of their decay phase (indicative of SERCA activity) between genotypes (**Figure 3e**). The dashed lines show single exponential fits which were used to determine the rate constants for the decay of $[Ca^{2+}]_i$, which was significantly slower by ~50 % in cells from KI mice (**Figure 3f**).”*

“Analysis of myocardium from WT or C42S PKG1 α KI mice—which cannot form the activating disulfide bond—revealed a significant deficit in PLN Ser16 phosphorylation in the mutant tissue, consistent with our biochemical data. This reduced phosphorylation PLN Ser16 produces substantial alterations in Ca²⁺ handling. PLN binds to SERCA, the pump that mediates Ca²⁺ reuptake in the SR, and reduces its activity. Therefore, SERCA activity influences the rate of decay of the Ca²⁺ transient and the amount of Ca²⁺ stored in the SR (SR Ca²⁺ content). The latter controls the amplitude of the Ca²⁺ transient therefore SERCA activity also controls the amplitude of the Ca²⁺ transient. PLN Ser16 phosphorylation relieves inhibition of SERCA2a resulting in increased activity, hastening the rate of decay of the Ca²⁺ transient, increasing both the SR Ca²⁺ content the Ca²⁺ transient amplitude. On the basis of these considerations it is not surprising to find that the lower levels of PLN Ser16 phosphorylation observed in the C42S PKG1 α KI are associated with a substantial reduction in the rate of decay of the Ca²⁺ transient, the SR Ca²⁺ content, as well as the Ca²⁺ transient amplitude. PLN can be phosphorylated at serine 16 by PKA following adrenergic stimulation. In addition, phosphorylation of threonine 17 by Ca²⁺/calmodulin-dependent kinase II (CaMKII) also increases SERCA activity¹⁵. The

main physiological function of CaMKII-mediated phosphorylation is to adapt SERCA function to increases in heart rate. Identification of PLN Ser16 as a selective target of oxidized PKG1 α raised a question about the functional significance and role of this phosphorylation event. We reasoned that oxidants produced during diastolic stretch (X-ROS signals)¹³ may trigger disulfide activation of PKG1 α and subsequent phosphorylation of PLN Ser16. This stretch-dependent myocardial oxidant signalling should be deficient in the hearts of the KI mice, because of the inability of C42S PKG1 α to transduce X-ROS signals into Ser16 PLN phosphorylation via the disulfide activation pathway. Indeed, we observed that cardiac stretch induced oxidation of PKG1 α in WT hearts, translocation of the kinase to the SR fraction, and an increase in PLN Ser16 phosphorylation. These events were absent in the KI hearts as hypothesized. On the basis of these observations, we speculated that the relationship between stretch and systolic contraction i.e., the Frank-Starling mechanism might be impaired in the KI hearts. Indeed, contractile responses were markedly impaired in isolated KI hearts compared to WT as EDP (stretch) was progressively increased up to 10 mmHg. This reduction in contractile responses is due both to impaired systolic function secondary to smaller Ca²⁺ transients because of the lower SR Ca²⁺ content, as well as diastolic dysfunction secondary to the slower Ca²⁺ transient decay, which will impair cardiac filling. These observations clearly delineate a novel role for the oxidised PKG1 α -PLN-SERCA axis in the modulation of the Frank-Starling mechanism, which traditionally has been attributed to modulation of myofilaments properties.

At this point it is important to highlight an apparent discrepancy in our data. The observation that the systolic Ca²⁺ transient was substantially lower in the KI cardiomyocytes compared to WT would be anticipated to manifest itself as attenuated cardiac systolic force development in vivo or in isolated heart preparations. However, this is not the case, as left ventricular systolic output measured by a PV catheter is identical between genotypes. This potential discrepancy can be fully reconciled by the fact that the KI is able to achieve the same systolic output as the WT in vivo at the cost of a sustained elevation in EDP. Similarly, isolated KI hearts can generate the same systolic pressure as the WT, but again this is only achieved by increasing their EDP above that required in WT. Thus at an EDP of 4 mmHg KI hearts only generate ~50% of the systolic output of a WT, matching very well the proportional deficit in the systolic Ca²⁺ amplitude measured in isolated unloaded cardiomyocytes. Essentially, the alterations to Ca²⁺ cycling observed in the KI reflect the inability to engage the Frank-Starling as effectively as WT myocardium. At very high EDP the preload will overcome the diastolic dysfunction and it is conceivable that additional mechanisms, not defined here, involving cardiac myofilaments or SR Ca²⁺ load adaptation participates to normalize systolic function.“

We have not performed the hydrogen peroxide treatments mentioned by the reviewer. Our rationale for not doing so is that X-ROS formation is unlikely to be modelled appropriately by application of exogenous hydrogen peroxide. Hydrogen peroxide is anticipated to target a great many proteins, whereas the production of oxidants during diastolic stretch is probably compartmentalised and generated at is sufficient to have localised effects.

In this connection, we know that many cardiac excitation-contraction coupling proteins, both ion translocators as well as myofilament proteins, are susceptible to direct regulation by oxidation. However, in our experience a major other confounding factor is that that other kinases that impact significantly on cardiac excitation-contraction coupling are also activated by hydrogen peroxide. For

example, there have been many studies showing that CaMKII is oxidised by hydrogen peroxide, which makes it constitutively active, as originally shown by Erickson and colleagues (Cell 2008). Although this oxidised and active CaMKII can target several excitation-contraction coupling proteins, it is notable that PLN threonine 17 (which is directly next to serine 16 which disulfide-activated *PKG1 α* phosphorylates) will become phosphorylated when cardiac myocytes are exposed to hydrogen peroxide. However, PLN threonine 17 is not phosphorylated by the subtler, likely compartmentalised oxidant production induced by stretch. Indeed, we show in the revised manuscript (Figure 1c) that PLN threonine 17 is not phosphorylated in response to diastolic stretch. Thus, we think that experiments with hydrogen peroxide will be limited by this issue. Furthermore, we have previously published (Brennan and colleagues, JBC 2006) on the oxidant-induced activation of Type I Protein Kinase A being mediated by oxidation of its regulatory RI subunit. This mechanism also induces phosphorylation of several excitation-contraction coupling proteins, including PLN serine 16. Again, this illustrates the complexities of using hydrogen peroxide and the limited conclusions that we would be able to be made, as other kinases are activated by this oxidant in the KI and these will phosphorylate targets directly relevant to cardiac contractile regulation. We therefore hope our rationale for not performing experiments with hydrogen peroxide are clear and their exclusion has been fully justified.

Reviewer #2 (expert in redox chemistry and cardiovascular disease)
Comments to the Author

--The authors should determine the phospholamban K_d for the disulfide form of PKG and for the C42S PKG mutant.

The K_d for the disulfide PKG α -PLN peptide interaction was determined by ITC and is now stated in the main results section:

“To explore the PLN-PKG α interaction further we carried out isothermal titration calorimetry (ITC), titrating the cytoplasmic domain of PLN (residues 1-23; PLN₁₋₂₃) against the oxidized (WT) and reduced (C42S mutant) forms of PKG α . A sigmoidal binding isotherm was fitted to the integrated titration data for oxidized PKG α which is consistent with one PKG α disulfide dimer binding to one PLN peptide with a K_d of $\sim 7 \mu\text{M}$ (Figure 2e). In contrast, integrated heats for the mutant kinase, recorded under the same experimental conditions, could not be fitted to a sigmoid-shaped binding curve, therefore a dissociation constant for the C42S PKG α -PLN₁₋₂₃ complex could not be derived from our experiments. Although the ITC data here does not exclude the possibility of an interaction between mutant PKG α and PLN, it does suggest that the interaction between reduced, unactivated PKG α and PLN is markedly weaker than the interaction between oxidant-activated PKG α and PLN. Using the MicroCal isotherm simulation tool, we estimated that the K_d for the mutant kinase is at least 5-fold higher than the K_d for WT PKG α disulfide dimer.”

As is stated clearly in the results section, our ITC experiments cannot rule out a direct interaction between the C42S mutant and phospholamban under the optimised experimental conditions that we used (70 μM protein in the cell and 700 μM peptide in the syringe). A simulation tool suggested a $K_d > 35 \mu\text{M}$, therefore we tried to obtain a binding affinity for this putative interaction by increasing the concentration of both protein and peptide. However, these experiments were hindered by protein aggregation and excessive bubbling in the ITC cell (a commonly encountered problem when studying weak protein-protein/peptide interactions that require high concentrations to be characterised experimentally).

Although we were unable to further characterise a direct mutant kinase-PLN interaction, we believe that the pertinent information relating to our study is that oxidation of PKG α promotes binding of the kinase to phospholamban. Our ITC data clearly shows this regardless of an absolute K_d value for the C42S PKG-PLN complex.

--What is the reversibility of the disulfide oxidation of cys42 in PKG?

We have evidence from numerous experiments that the Cys42 disulfide bond is reversible under physiological conditions.

We have published data showing the accumulation of PKG α disulfide dimer upon treatment of HEK cells, murine aortic rings or mesenteric blood vessels with the thioredoxin reductase inhibitor auranofin—consistent with the oxidized kinase being continually (even under basal conditions) redox recycled back to the reduced form by the cellular NADH-dependent thioredoxin reductase reducing system (Burgoyne et al. Hypertension 60, 1301-1308 (2012); Rudyk et al. Circulation 126, 287-295 (2012)). Moreover, we have previously found that a mixed disulfide forms between Cys35Ser of

thioredoxin and PKG after isolated hearts were exposed to hydrogen peroxide, which is fully consistent with thioredoxin reducing disulfide PKG back to its reduced state. These observations were made in hearts from a thioredoxin 'trap mutant' transgenic mice expressing FLAG-C32S Trx-HA, which allows proteins reductively recycled by thioredoxin in the heart to be identified. This transgenic model was developed by the Sadoshima lab (Newark, USA) and the data are included in a PhD thesis (2015) by Daniel Stubbert from this lab. We have amended the manuscript so that it now states that the PKGI α Cys42 disulfide bond is reversible and referenced the literature detailing the auranofin experiments described above:

“Reactive oxygen species (ROS) promote formation of a reversible intermolecular disulfide bond between the two subunits of the PKGI α homodimer at Cys42^{4,5}.”

Furthermore, in relation to reversibility, we also have unpublished data relating to the thermodynamic stability of the Cys42 interchain disulfide bond *in vitro*. We have performed a dihydrolipoic acid/lipoic acid redox titration with recombinant PKGI α and analysed the amount of disulfide dimerization by SDS-PAGE. The equilibrium redox potential was determined as approximately -280 mV; a value within the confines of the cellular redox potential which ranges from ~ -350 to -150 mV depending on e.g. the organelle and stage of the cell cycle (Jones, D. P. *Journal of Internal Medicine* 268, 432-448 (2010); Jones, D. P. *American Journal of Physiology - Cell Physiology* 295, C849-C868 (2008)). This suggests that PKGI α is well poised to become oxidized under physiological conditions yet still able to be efficiently reduced by the cellular reducing systems—consistent with our findings from tissue and cells, and arguably exactly what would be expected of a protein that is required to rapidly transduce an oxidant signal into a biological response.

--What are the consequences of cys42 oxidation for cGMP activation of PKG?

Müller *et al* found that pre-oxidation of PKGI α with hydrogen peroxide slightly attenuated its activation by cGMP in mouse embryonic fibroblasts, inferred from phosphorylation of a known substrate, VASP (Muller *et al.* *Free Radic Biol Med* 53, 1574-1583 (2012)). It is currently unknown if the diminished ability of PKGI α to phosphorylate VASP was due to reduced affinity of the kinase for cGMP, or whether oxidation in the N-terminal leucine zipper region inhibited kinase-substrate interactions; both of these mechanisms are entirely plausible. Indeed, the N-terminal leucine zipper domain of PKG (where Cys42 is located) is known to regulate the kinase's affinity for cGMP (Francis, D. *et al.* *Pharmacol. Rev.* 62, 525-563 (2010)) and is also important for substrate recognition (Lee, E., Hayes, D. B., Langsetmo, K., Sundberg, E. J. & Tao, T. C. *J. Mol. Biol.* 373, 1198-1212 (2007); Kato, M. *et al.* *J. Biol. Chem.* 287, 41342-41351 (2012)).

For the reverse situation—the effect of cGMP activation on PKGI α oxidation—we and others have shown that cyclic nucleotide binding attenuates formation of the Cys42 disulfide bond (Muller *et al.* *Free Radic Biol Med* 53, 1574-1583 (2012); Burgoyne *et al.* *Hypertension* 60, 1301-1308 (2012)). Interestingly, Alverdi *et al* found that the conformation of the N-terminal region of PKGI α is altered upon cGMP binding (*J. Mol. Biol.* 375, 1380-1393 (2008)); this structural change probably results in the two Cys42 residues becoming less susceptible to oxidation.

Overall the evidence that has accumulated to date therefore suggests that there is a degree of mutual exclusivity associated with the two activation modes of PKGI α . This is consistent with PKGI α taking part in discrete signalling pathways that diverge depending on the stimulus. One mode of

PKGI α activation blocking the other would ensure that the selectively-activated kinase could fulfil its downstream function without interference from an alternative pathway that could potentially be stimulated at the same time.

We have included some content related to the reviewer's query in the Discussion:

"PKGI α activity can be stimulated by reversible oxidation of Cys42 or by cGMP binding^{6,7}. cGMP inhibits formation of the Cys42 intermolecular disulfide bond^{4,5} and, similarly, oxidation has been shown to attenuate cGMP-dependent PKGI α substrate phosphorylation⁸, consistent with discrete PKGI α signalling pathways that diverge depending on the stimulus."

--The authors should perform formal P-V loop analysis of the LV for the WT and C42S PKG mutant.

We did in fact perform conventional PV loop analyses for the LV of the WT and mutant mice, and we discussed this data in the main text (differences in EDP, relaxation and contraction rates, ESPVR, EDPVR, and Tau were described) but perhaps it was not clear enough because these results were included in the same paragraph as the *in vivo* echocardiography data in the original submission:

"In vivo echocardiography showed the KI mice, which are resistant to PKGI α oxidative activation, have a lower E/A ratio than WTs, indicating impaired diastolic relaxation (Figure 3a). An additional in vivo comparison of cardiac performance by PV loop analysis corroborated this diastolic dysfunction; the KI hearts displayed increased EDPs and were slower in both their contraction and relaxation rates (Figure 3b). This observation is consistent with the ex vivo isolated heart preparations described above. The reduced end-systolic pressure-volume relationship (ESPVR) in the KI shows a deficiency in contractile performance, and the elevated end-diastolic pressure-volume relationship (EDPVR) indicates a stiffer ventricle (Figure 3b). Impaired relaxation in the KI was further supported by an increased isovolumic relaxation constant, Tau (Figure 3b), providing a preload-independent measure of diastolic function."

We have amended the manuscript so that the standard PV loop data is described in its own paragraph (prior to any mention of the novel analysis, which we describe under the next subheading). In addition to the histograms displayed in Figure 4b, which are derived from the conventional PV loop analyses, we have also added a table documenting the complete list of standard PV loop measurements into the Supplementary Information:

"In vivo cardiac performance was further assessed by analysis of pressure-volume (PV) loops obtained from a catheter inserted in the left ventricle (LV) of the WT and KI mice; the complete list of measurements can be found in Supplementary Table 2. The KI hearts had significantly increased EDPs and were slower in both their contraction and relaxation rates (Figure 4b). Furthermore, the reduced end-systolic pressure-volume relationship (ESPVR) in the KI reveals a deficiency in contractile performance, and the elevated end-diastolic pressure-volume relationship (EDPVR) indicate that the ventricle of the KI is stiffer. Compelling evidence for diastolic impairment in the KI is further provided by an elevated isovolumic relaxation constant, Tau, which is a preload-independent measure of diastolic function; a higher value indicates slower relaxation²²."

--The authors do not consider other potential molecular mechanisms that may in part account for the Frank-Starling effect, including phospholamban S-nitrosylation (cf. Garofalo F et al., Proc Biol Sci 2009;276:4043-52-in fish) or cardiac myosin-binding protein C and Tnl phosphorylation (Kumar M et

al., JBC 2015;290:29241-9), to name but two. How does PKG cys42 oxidation compare to these potential regulatory mechanisms in governing the overall phenomenon?

The manuscript was previously formatted for a different journal and we were constrained by the word limit. However, we agree with the reviewer that the manuscript would benefit from more discussion of other cellular mechanisms that might also account for the diminished Frank-Starling response of the KI heart. We have changed the format of the article to adhere to Nature Communications guidelines and been able to address in more detail underlying mechanisms that are responsible for the altered cardiac function of the KI *due* to under phosphorylation of PLN Ser16 (i.e. dysfunction as a result of impaired Ca^{2+} handling and changes in myofibril Ca^{2+} sensitivity because of inadequate stretching of the sarcomeres) and highlight some other potential mechanisms that might occur independently of PLN phosphorylation status (i.e. phosphorylation of other targets such as those mentioned by the reviewer):

“Deficiencies in the cardiac response to preload in the C42S PKG α KI mice are due, at least in part, to insufficient phosphorylation of PLN Ser16 during ventricular filling. The resulting elevation in diastolic Ca^{2+} concentration means that the myocardium cannot relax properly and a reduced SR Ca^{2+} load also leads to diminished contractility³⁵. Passive tension is also likely to play a role in the diminished Frank-Starling responses of the KI hearts due to increased interactions of the giant elastic protein titin with Ca^{2+} . Ca^{2+} binding to titin is known to increase passive tension of the myocardium, making the ventricle stiffer and thus harder to fill¹². Furthermore, because impaired relaxation results in inadequate extension of the sarcomeres, the myofilament Ca^{2+} sensitivity—which is dependent on sarcomere length^{11,12,29}—will be reduced in the KI cardiomyocytes and also contribute to the decreased contractility observed in the KI hearts^{11,12,36}.

Other molecular mechanisms that facilitate Frank-Starling responses may also be altered in the absence of the PKG α oxidation pathway. For example, phosphorylation of the sarcomeric proteins cTnI and cMyBP-C is involved in length-dependent activation of myofilaments³⁷. However, we did not observe changes in phosphorylation of key residues in these proteins in the KI hearts. Basal phosphorylation of several other phosphosites involved in cardiac EC coupling and Ca^{2+} handling including phospholemman Ser68 and CaMK2 Thr282 was also unchanged in the KI hearts, suggesting that these residues are not central to the X-ROS/PKG α oxidation/enhanced myocardial relaxation pathway. S-nitrosylation of PLN is another modification that has been shown to modulate the Frank-Starling mechanism and it is possible that this modification has a role in X-ROS signalling³⁸. However, we cannot envisage how the PKG α Cys42Ser mutation could affect PLN S-nitrosylation and it is unlikely that this redox modification contributes to the diminished Frank-Starling mechanism observed in the KI mice.”

Reviewer #3 (expert in phospholamban)

Comments to the Author

The paper by Scotcher et al report a chemical screen showing that PKG targets phospholamban, that PKG KI mice have reduced Frank Starling responses, and conclude that the impaired response is due to a phosphorylation of PLN.

There are aspects of this study that have merit and are novel. But, as presented, this manuscript is not acceptable for publication.

The data in the PKG KI mice by isolated in vitro hemodynamics is clear (Fig 2a,b,d,e).

Major concerns

1. The potential role of PKG in regulating PLN to contribute to Frank Starling responses would be interesting. However, challenging the conventional idea that the response is primarily due to tension-length aspects, rather it is due to direct phosphorylation status MUST be proven. The authors made no effort to document myosin length measures, nor phosphorylation status of myosin! This would be a critical fatal flaw that would need to be addressed.

To clarify this issue, we are not trying to challenge the well-established and widely accepted mechanical basis of the Frank-Starling mechanism in any way i.e. increased overlap of the myofilaments, decreased myofilament lattice spacing, increased cross-bridge formation and increased Ca^{2+} sensitivity. In fact, we think that PKGI α -dependent phosphorylation of PLN contributes to the Frank-Starling length-tension relationship by enabling enhanced stretch of the sarcomeres via increased Ca^{2+} uptake into the SR during diastole. It is already known that changes in preload alter calcium handling, and a clear distinction between purely mechanical aspects of the Frank-Starling mechanism and contributions from signalling via post-translational modifications cannot be made. There are numerous underlying components that facilitate enhanced cardiac output on a beat-to-beat basis as venous return is increased, and we describe a novel one in our manuscript. Stretch-dependent phosphorylation of PLN via oxidation of PKGI α seems to be a mechanism which links mechanical changes with altered calcium handling.

We are pleased that the reviewer thinks that the novel role of PKG in regulating PLN to contribute to the Frank-Starling mechanism is interesting, because this is exactly what we describe in the manuscript. We show conclusively that when the PKGI α oxidation pathway is absent, as is the case in the C42S PKGI α KI mouse, the Frank-Starling response is diminished. Impaired Ca^{2+} handling accounts for the reduced cardiac performance of the KI hearts, at least in part, because PLN Ser16 cannot be phosphorylated by disulfide-activated PKGI α . Thus, in the model we present evidence for, the X-ROS mechanism is primarily important to the Frank-Starling mechanism by ensuring that the heart relaxes adequately during diastole, ensuring adequate extension of the sarcomeres. Thereafter, the myofilament-dependent basis of the Frank-Starling mechanism remains unchanged, and so monitoring aspects like myosin phosphorylation are not directly relevant. To reiterate, the KI has an impaired Frank-Starling response because it doesn't relax adequately because it cannot couple X-ROS levels to activation of SERCA2a, and thus what might be considered the 'start' of the Frank-Starling mechanism that draws the myofilaments apart is not engaged as it is in WT.

By using the word “mediates” in the original title of the manuscript “*Oxidation of protein kinase G mediates the Frank-Starling law of the heart in vivo*” perhaps gives the unintended impression that we believe PKGI α oxidation is the primary underlying mechanism of the Frank-Starling response. Thus, we have changed it to “*Disulfide-activated protein kinase G regulates diastolic relaxation of the heart and fine-tunes the Frank-Starling response*” which the reviewer will hopefully find a more accurate reflection of the data and conclusions. As the reviewer mentions in a comment below, formatting of the manuscript made it difficult to read. The manuscript has been extensively revised and the data and subsequent conclusions should be much less open to misinterpretation now.

2. For control studies in the PKG KI mouse, the authors must provide additional controls, not just examine the status of PLN. I fully appreciate they identified one peptide for PLN-pS and that is their focus. However, RyR, DHPR, CamKII, myosin heavy chain, as examples, MUST be examined with respect to their total levels and their phosphorylation status. This would be a basic fundamental requirement.

We have assessed the total levels and phosphorylation status of multiple proteins involved in calcium handling and length-dependent activation of cardiomyocytes and added these data into Figure 1c. For the proteins and phosphosites we were able to detect with antibodies that ‘worked’ in our hands, we observed no changes in either total levels or phosphorylation in the C42S PKGI α KI compared to WT. Unfortunately, we are yet to obtain a commercial phospho-DHPR antibody that works in our hands. Also, as far as we are aware there is very little known about phosphorylation of cardiac myosin heavy chain nor any commercial antibodies for a phosphosite available, but we realise that the levels of myosin heavy chain isoforms (alpha versus beta) can modulate the Frank-Starling response (Korte FS, McDonald KS. *J Physiol.* J581(Pt 2):725-39 (2007)), if they were altered in the KI. Consequently, we measured and compared the levels of these isoforms of myosin heavy chain in WT and KI, but found no expression difference between genotypes, and so an isoform switch cannot account for the impaired the Frank-Starling response of the transgenic. Results from these additional control experiments suggest that the other key residues we analysed are not important in the X-ROS/PKGI α oxidation/myocardial relaxation pathway that we describe in the manuscript.

We amended the text to include the control experiments as follows:

“Given that PLN plays a central role in cardiac excitation-contraction (EC) coupling and Ca²⁺ homeostasis, we analysed several other key proteins involved in these processes to see if their phosphorylation status was also altered in the KI myocardium (Figure 1c). However, we observed no changes at any of the phosphosites we analysed: cardiac troponin I (cTnI) Ser22/23, cardiac myosin binding protein C (cMyBP-C) Ser282, ryanodine receptor 2 (RyR2) Ser2808, phospholemman (FXYP1) Ser63, Ser68, and Ser69, myosin light chain 2 (MLC2) Ser19, and Ca²⁺/calmodulin-dependent protein kinase II (CaMK2- $\beta/\gamma/\delta$) Thr282.”

“Other molecular mechanisms that facilitate Frank-Starling responses may also be altered in the absence of the PKGI α oxidation pathway. For example, phosphorylation of the sarcomeric proteins cTnI and cMyBP-C is involved in length-dependent activation of myofilaments³⁷. However, we did not observe changes in phosphorylation of key residues in these proteins in the KI hearts. Basal phosphorylation of several other phosphosites involved in cardiac EC coupling and Ca²⁺ handling including phospholemman Ser68 and CaMK2 Thr282 was also unchanged in the KI hearts, suggesting that these residues are not central to the X-ROS/PKGI α oxidation/enhanced myocardial relaxation pathway. S-nitrosylation of PLN is another modification that has been shown to modulate the Frank-

Starling mechanism and it is possible that this modification has a role in X-ROS signalling³⁸. However, we cannot envisage how the PKG α Cys42Ser mutation could affect PLN S-nitrosylation and it is unlikely that this redox modification contributes to the diminished Frank-Starling mechanism observed in the KI mice.”

3. The final figure 3g has absolutely no supporting data in their paper. In fact, the authors concluded "Scheme showing how oxidation of PKG α by stretch induced oxidants mediates the Frank-Starling response" This paper presents no direct evidence that ROS levels were elevated in their model, nor that SERCA activity was increased. To make these conclusions, the authors would need to measure ROS levels in their systems, perhaps scavenge ROS levels, assess Ca²⁺ store levels, assess Ca²⁺ transients, examine responses with/without ISO treatment.

We have amended the caption for Figure 4g:

“Scheme showing how oxidation of PKG α by stretch-induced oxidants contributes to regulation of the Frank-Starling response.”

We think that PKG oxidation in itself provides a valid readout of ROS in the system. The likelihood is that the X-ROS generated are localised and generated at relatively low levels that would make it extremely challenging to measure using conventional biochemistry. Because of the anticipated changes in ROS during myocardial stretch are likely localised and subtle, this would allow selective redox signalling that avoids cell-wide changes in redox state, which would cause many other redox-active proteins being modulated – which is probably not a useful feature for a regulatory mechanism. Thus to reiterate, this subtle and localisation means that measurements of oxidants in the bulk tissue will not detect the oxidation that we have observed by monitoring PKG oxidation.

The reviewer mentioned ROS scavenging (i.e. antioxidant) treatment, which many would perhaps expect to attenuate oxidation of PKG. We realise that this concept is found extremely widely in the literature, and indeed many studies use such interventions to achieve the hypothesized result, namely blockade of target oxidation. This idea is essentially the same as antioxidants being therapeutic by preventing target oxidation. However, we should remember that although many preclinical studies routinely show antioxidant working in this way, that human clinical trials have failed. Indeed, in many cases the antioxidant interventions actually caused harm with many trials being stopped early because of increased mortality. This has called into question the typical blockage of oxidation and protection shown by multitudes of preclinical studies. Antioxidants are reductants, which by definition are electron donors, and so potentially may fuel oxidase enzymes to actually enhance oxidant generation and exacerbate target oxidation. Antioxidants such as N-acetylcysteine (NAC), which may be the compound of choice for many in a study like ours, will react with oxidants to generate a disulphide derivative, which can potentially induce protein oxidation via thiol–disulphide exchange chemistry – likely triggering PKG oxidation. Another major issue with antioxidants like NAC, is that many oxidants react faster with target protein thiols than with the antioxidant thiol, so even when these small reducing molecules are used at high concentration they may not prevent target oxidation. Even if antioxidants effectively intercept oxidants, eventually they may become exhausted and, as mentioned, their oxidized end-products which themselves may induce protein oxidation accumulate. Thus antioxidants are likely not as effective, in the context of blocking oxidant-induced signalling, as many may think and because of our experience with them in the thiol redox field, we try to avoid their use. This is because the results obtained with them, for the

reasons explained, are complicated to interpret and they are not anticipated to be as effective as many would envisage. Indeed, we used NAC to try and block oxidation of another kinase (www.nature.com/ncomms/2015/150810/ncomms8920/full/ncomms8920.html), and this actually promoted the oxidation of that target and exacerbated the pro-cancer signalling that it would supposedly attenuate. Several large trials testing if antioxidants protect against cancer have been stopped – this is because the antioxidants increased mortality very significantly. Now high profile preclinical studies show that antioxidants promote cancer (<http://stm.sciencemag.org/content/7/308/308re8>, www.nature.com/nature/journal/v475/n7354/full/nature10189.html) and indeed giving oxidants can be therapeutic (www.nature.com/nature/journal/v527/n7577/full/nature15726.html). The main point we are making is that it is not especially helpful or illuminating to perform such antioxidants studies.

We have now included the SR calcium data mentioned by the reviewer, as part of wider measurements on calcium cyclin in isolated adult ventricular cardiomyocytes from WT or KI hearts. We have included the following results text to the manuscript in direct relation to this point:

*“Experiments were performed in ventricular myocytes isolated from adult WT or C42S PKG1 α KI hearts, comparing intracellular calcium ($[Ca^{2+}]_i$) dynamics between genotypes. Specimen transients (**Figure 3a**) are clearly consistent with significantly altered Ca^{2+} handling in the cells from KI animals. Quantitative analysis of the transients showed the KI was significantly deficient in their systolic $[Ca^{2+}]_i$ transient and SR Ca^{2+} content evoked by application of caffeine, whereas the diastolic $[Ca^{2+}]_i$ concentration was the same between genotypes (**Figure 3b-d**). Normalisation of the $[Ca^{2+}]_i$ transients allowed direct comparison of their decay phase (indicative of SERCA activity) between genotypes (**Figure 3e**). The dashed lines show single exponential fits which were used to determine the rate constants for the decay of $[Ca^{2+}]_i$, which was significantly slower by ~50 % in cells from KI mice (**Figure 3f**).”*

Minor concerns

-Formatting of this manuscript made it extremely difficult to read. For instance, there is no introduction. Rather the paper goes from a general abstract directly into an Intro/Results/Discussion. Figure legends are not conventional, rather an expanded results section. Guidelines for NComm submission are online and very clear. As a result, the authors had an opportunity to frame their discovery in perspective but failed to do so.

Apologies to the reviewer – the manuscript was previously formatted for a different journal where we were very constrained by the word limit and it was transferred directly to Nature Communications via an automated service without the appropriate revisions. The manuscript has been rewritten to adhere to Nature Communications guidelines; there is now a designated introduction, results, and discussion section, the Figure legends are much more concise, and we have been able to interpret our results and relate them to the existing literature more comprehensively.

Echo data in Figure 3 a/b might be better presented as a standard table, as in most studies.

A lot of *in vivo* data is compiled in Figure 4 to specifically show that diastolic relaxation and the Frank-Starling mechanism are impaired in the KI mice. For brevity we decided to clearly show only the most pertinent echocardiography data in this Figure (i.e., the *E/A* and *E'/A'* ratios which are well established clinical markers of diastolic function). The full list of echocardiography measurements is presented in a standard format in Supplementary Table 1 as requested by the reviewer.

-Fig 1b. were these blots boiled, what ratio of pentamer/monomer was documented?

Samples for PLN immunoblots were immediately boiled prior to gel loading as is stated in the Methods section. However, we have now added data from immunoblots for samples that were not boiled into Figure 1b and amended the results section and discussion text accordingly:

“As well as Ser16 phosphorylation, the oligomeric state of PLN was altered in the myocardium of the KI, as indicated by a 3-fold increase in the pentamer/monomer ratio of total PLN in samples that were not boiled prior to Western blotting.”

“We observed an increase in the pentamer/monomer ratio of PLN in the KI myocardium compared to the WT, which may represent an adaptive change in the KI. However, it is difficult to make firm conclusions about the significance of this observation, as there is evidence for^{16, 30}, as well as against³¹, oligomerization mediating activation of SERCA2a. We should also consider that there are other regulatory mechanisms that participate in the control of SERCA2a activity, such as dephosphorylation of PLN by protein phosphatase 1, and by differential interactions with SR membrane proteins including sarcolipin and DWORF^{32, 33, 34}.”

REVIEWERS' COMMENTS:

Reviewer #1 (Remarks to the Author):

This referee did not submit comments to the author, but in her/his confidential comments to the editor confirms that the conclusions are now validated by the study and supports the study's publication.

Reviewer #2 (Remarks to the Author):

I have no comments for the authors.

Reviewer #3 (Remarks to the Author):

This revised manuscript addressed my major concerns in a satisfactory way. The greater level of careful interpretation and discussion of the data is now appropriate.

I have no further comments or concerns for this manuscript.